# Berberine Regulation of Cellular Oxidative Stress, Apoptosis and Autophagy by Modulation of m^6^A mRNA Methylation through Targeting the *Camk1db*/ERK Pathway in Zebrafish-Hepatocytes

**DOI:** 10.3390/antiox11122370

**Published:** 2022-11-30

**Authors:** Meijuan Zhang, Jin Liu, Chengbing Yu, Shangshang Tang, Guangzhen Jiang, Jing Zhang, Hongcai Zhang, Jianxiong Xu, Weina Xu

**Affiliations:** 1Shanghai Key Laboratory for Veterinary and Biotechnology, School of Agriculture and Biology, Shanghai Jiao Tong University, No. 800 Jiangchuan Road, Shanghai 200240, China; 2Key Laboratory of Aquatic Nutrition and Feed Science of Jiangsu Province, College of Animal Science and Technology, Nanjing Agricultural University, No. 1 Weigang Road, Nanjing 210095, China

**Keywords:** berberine, m^6^A methylation, *Camk1db*, ERK, zebrafish hepatocytes (ZFL)

## Abstract

Berberine (BBR) ameliorates cellular oxidative stress, apoptosis and autophagy induced by lipid metabolism disorder, however, the molecular mechanism associated with it is not well known. To study the mechanism, we started with m^6^A methylation modification to investigate its role in lipid deposition zebrafish hepatocytes (ZFL). The results showed that BBR could change the cellular m^6^A RNA methylation level, increase m^6^A levels of *Camk1db* gene transcript and alter *Camk1db* gene mRNA expression. Via knockdown of the *Camk1db* gene, *Camk1db* could promote cellular ERK phosphorylation levels. Berberine regulated the expression level of *Camk1db* mRNA by altering the M^6^A RNA methylation of the Camk1db gene, which further affected the synthesis of calmodulin-dependent protein kinase and activated ERK signaling pathway resulting in changes in downstream physiological indicators including ROS production, cell proliferation, apoptosis and autophagy. In conclusion, berberine could regulate cellular oxidative stress, apoptosis and autophagy by mediating *Camk1db* m^6^A methylation through the targeting of the *Camk1db*/ERK pathway in zebrafish-hepatocyte.

## 1. Introduction

The long-term excessive exposure to a high-fat diet results in lipid metabolism disorders and over-production of reactive oxygen species (ROS), which triggers oxidative stress and cell damage at the cellular and molecular levels [1]. Studies have demonstrated cell damage biochemical mechanisms involved in the cellular signaling pathways [2] and the interaction between genes and protein expression changes at the transcriptional level [3,4]. Gene RNA is a key linking DNA to protein in the transmission of genetic information, but the level of synthesized protein does not necessarily correlate positively with the level of mRNA, suggesting the importance of post-transcriptional RNA modifications. There are more than 100 known post-transcriptional modifications of RNA. Among these, m^6^A methylation is the most common type [5], and also the most widely studied dynamic modification method of RNA. M^6^A RNA methylation, the sixth methylation modification of adenine on nitrogen atoms, is an evolutionarily-conserved RNA modification that is present in most organisms from bacteria to mammals [6,7]. M^6^A RNA methylation is also a reversible epigenetic RNA modification that exists at the transcriptional level of the nitrogen or oxygen atom of S-adenosylmethionine, and regulates mRNA stability, splicing, export, localization, and translation [8,9,10]. M^6^A RNA methylation can affect most aspects of gene expression. Recent studies have identified m^6^A RNA methylation modifications plays an important role in regulating cell biological function, controlling cell proliferation and differentiation [11], cellular oxidative stress response [12,13], DNA damage response [14], and lipid metabolism [15,16,17,18]. Now, biological functions for this m^6^A modification are emerging through more mechanistic analyses. 

The Chinese medicine berberine (BBR, chemical formula C_20_H_18_NO_4_) is an isoquinoline quaternary alkaloid isolated from several medicinal plants, including *Berberis aristata* and *Coptis chinensis* [19,20]. Berberine can inactivate highly active molecules such as O_2_^•−^, HO^•^, NO and OONO^−^, so as to facilitate the ability to remove reactive oxygen species clusters (ROS) directly [21]. In addition, modern pharmacological studies have shown that berberine exhibits hepatoprotective, anti-inflammatory, antioxidant, hypoglycemic, hypolipidemic properties [22,23], and reduces obesity in mammals [24]. Our previous studies showed berberine-supplemented diets could attenuate oxidative stress, improve the function of the mitochondrial respiratory chain, reduce apoptosis and inhibit the inflammation response as well as modulate the intestinal microflora profile caused by high-fat and high-carbohydrate diets in fish in vitro and vivo [25,26,27,28]. However, the molecular mechanism of berberine’s role in oxidative stress, apoptosis, and autophagy is still limited. In this paper, we investigated whether berberine could regulate cellular oxidative stress, apoptosis and autophagy by changing the methylation level of cell gene m6A RNA.

Zebrafish (*Danio rerio*, Cyprinidae) is an important vertebrate model organism for development, genetics, and reproduction [29]. The liver is the main organ for fat synthesis, and about 90% of the fat in fish is synthesized in the liver [30], which plays an important role in maintaining the metabolic balance of carbohydrates, amino acids, and fatty acids [31]. There were many studies using Zebrafish hepatocytes (ZFLs) to study fat deposition in vitro. Therefore, in this study, we selected ZFL as model system for the study of metabolic syndrome combined with oxidative stress to investigated the roles of m^6^A RNA methylation modification in lipid-deposition ZFL treated with berberine in vitro. The results may have implications for a molecular theoretical basis for the research and development of hepatoprotective herbal treatments in fish farming.

## 2. Materials and Methods

### 2.1. Materials

Zebrafish hepatocytes (ZFL, CRL-2643) were purchased from the American Tissue Culture Collection (ATCC, Manassas, VA, USA). Other materials and reagents purchased included the following: DMEM-F12 medium, DMEM medium, L-15 medium, fetal bovine serum (Gibco, Waltham, MA, USA); penicillin, murine epithelial growth factor, insulin, DPBS and 0.25% trypsin (Thermo Fisher, Shanghai, China); total RNA extraction kit (OMEGA, Cambridge, MA, USA); reverse transcription kit, and fluorescent quantitative PCR kit (Takara, Beijing, China); the targeted antibody (ERK, P-ERK, P62, LC3-B) and the internal antibody tubulin (because the size of the antibody protein of interest is between 10 and 70 kDa, in order to ensure that the target internal reference is on a membrane, and our sample is a whole cell protein, the loading amount is 30 micrograms, using ripa lysate, so for comprehensive consideration, tubulin was selected as the internal reference protein) are purchased from CST (Parsons, KS, USA). Instrumentation included the following: high-speed frozen centrifuge (Eppendorf, Hamburg, Germany, 5804 R, Shanghai, China); real-time fluorescence quantitative PCR instrument (Applied Biosystems, StepOnePlus^TM^, Waltham, MA, USA); inverted microscope (Nikon Corporation, Hong Kong, China); multifunctional enzyme labeling instrument (SYNERGY 2 BioTek, Waltham, MA, USA). Oil Red O staining solution, sodium palmitate, and berberine, were purchased from Sigma-Aldrich (Shanghai, China) Trading Co. CCK-8 cell activity assay kit, BCA protein assay kit, triglyceride assay kit, ROS, MDA, and T-AOC assay kits, were purchased from Nanjing Jiancheng Institute of Biological Engineering. Total RNA extraction kit was purchased from OMEGA (Cambridge, MA, USA). Primers were synthesized by Thermo Fisher Scientific China Ltd (Shanghai, China). The reverse transcription kit PrimeScript RT reagent Kit (Perfect Real Time) and its matching fluorescent quantitative PCR kit TB Green^TM^ Premix Ex Taq^TM^ (Tli RNaseH Plus) were purchased from Takara Corporation (Beijing, China).

### 2.2. Lipid Accumulation Injury Treatment and Berberine Repair Treatment in ZFL

#### 2.2.1. Lipid Accumulation Injury Treatment in ZFL

Mixed media was composed of 50% L-15, 30% DMEM/F12, and 20% DMEM (containing 5% fetal bovine serum, 1% penicillin, 1% insulin and 50 ng/mL murine epithelial growth factor). ZFL cells were cultured to 70–80% fusion with a mixed culture base at 28°C and 5% CO_2_. Cells were collected after treatment with sodium palmitate (SP) culture solution at a concentration of 0.25 mmol/L for 24 h, thus establishing a validated model of hepatocyte fat deposition [32].

#### 2.2.2. Berberine Repair Treatment in ZFL

ZFL cells were treated with SP culture solution at a concentration of 0.25 mmol/L for 24 h and then collected. The cells were then treated with BBR working solution (solvent 0.1% DMSO) at a concentration of 25 µmol/L for 6 h and then collected, thus establishing an effective model of hepatocyte fat deposition repair [32].

### 2.3. Oxidative Stress Indicator Test

#### 2.3.1. ROS Content Detection

The intracellular ROS (reactive oxygen species) content was detected using a DCFH-DA probe. After each group of cells was treated, cells were rinsed with DPBS three times, incubated with DCFH-DA probe for 30 min, and rinsed with DPBS three times, and ROS fluorescence photographs were taken with fluorescence microscopy. Cells received the same treatment as above, with a microplate reader to determine the fluorescence intensity at an excitation wavelength of 488 nm and an emission wavelength of 525 nm. The calculation formula is as follows:$$\mathrm{ROS}\;\mathrm{content}=\left(\frac{\mathrm{Fluorescence}\;\mathrm{intensity}\;\mathrm{of}\;\mathrm{the}\;\mathrm{treatment}\;\mathrm{group}-\mathrm{Fluorescence}\;\mathrm{intensity}\;\mathrm{of}\;\mathrm{blank}\;\mathrm{group}}{\mathrm{Fluorescence}\;\mathrm{intensity}\;\mathrm{of}\;\mathrm{the}\;\mathrm{control}\;\mathrm{group}-\mathrm{Fluorescence}\;\mathrm{intensity}\;\mathrm{of}\;\mathrm{blank}\;\mathrm{group}}\right)÷\mathrm{Cell}\;\mathrm{density}\times 100\%$$

#### 2.3.2. MDA and T-AOC Contents Measurement

After the cell culture treatment, cells were placed on ice. The cell culture medium was discarded, rinsed 3 times with DPBS, and then 200 µL of cell lysate was added to each well for lysis. The cell lysate was collected, and the intracellular MDA and T-AOC contents were measured separately according to the method of the kit (Nanjing Jiancheng Institute of Biological Engineering).

### 2.4. Cell Apoptosis

Apoptotic cells were detected by Annexin V/FITC as described previously [33]. Each group of cells was collected in a centrifuge tube and centrifuged at 1000× *g* for 5 min. Supernatant was discarded and 195 µL of membrane linked protein V-FITC binding solution was added to gently suspend the cells. Next, 5 µL of Annexin V-FITC were added and mixed well before further adding 10 µL of propyl iodide solution. The mixture was incubated at room temperature for 10–20 min on ice, protected from light. Apoptosis was detected using flow cytometry counting, where the apoptosis rate (%) = number of apoptotic cells/total number of cells × 100%.

### 2.5. Western Blot

Cells were collected by centrifugation, lysed on ice for 30 min with RIPA cell lysis solution, and the total protein concentration was determined by BCA method. Each well was sampled with 30 µg of protein and separated by 10% polyacrylamide gel electrophoresis. The separated proteins were electro transferred (200 mA current for 2 h) to PVDF membrane. ERK (cst, 9102), P-ERK (cst, 9101), P62 (cst, 5114), LC3-B (cst, 2775) (1:1000 volume dilution) were taken from the primary antibody rabbit, and the internal reference primary antibody tubulin (cst, 2146) (1:2000 volume dilution) was taken, and incubated overnight at 4 °C. The next day, the membranes were washed three times with TBST for 10 min each time, followed by the addition of secondary antibodies against rabbit anti-IgG (1:2000 *v*/*v*) for 2–4 h at room temperature, and then washed three times with TBST for 10 min each time, and developed according to the instructions of the ECL kit. The grayscale values of the protein bands were analyzed using the LK5100 electrochemiluminescence analysis system.

### 2.6. Cellular Microstructure Detection

A sufficient number of cells were cultured in a 10 cm dish. Cells grown in the petri dish were scraped off directly and the cell suspension was transferred to the centrifuge tube. Cells were centrifuged at 1000–3000 rpm, the supernatant was discarded, and PBS was added to transfer the cells to a 1.5 mL centrifuge tube. After centrifugation, supernatant was discarded. PBS was added to repeat the above steps, and 2.5% glutaraldehyde fixative for tissue and cell electron microscopy was added at 4 °C overnight. Samples were then sent to Wuhan Max Bio-Technology Co., Ltd. (Wuhan, China) the next day for subsequent electron microscopy detection.

### 2.7. m^6^A Methylation and Transcriptome Merip-Seq Sequencing

The RNA of the total samples was isolated and purified using TRizol, and the amount and purity of the total RNA was tested for quality control and integrity, while the protocol was verified by agar electrophoresis. Magnetic beads with Poly adenylic acid were used to specifically capture the mRNA. Fragmentation was performed with a magnesium ion interruption kit, and the fragmented RNA was pre-mixed with immunomagnetic beads with m^6^A antibody, and subjected to IP. The IP product was subjected to duplex synthesis, converting the DNA and RNA duplexes into DNA duplexes, and complementing the ends. Base A was added at each end, ligated to the end with base T, and purified by screening with magnetic beads. After digestion of the duplexes, PCR was pre-denatured at 95 °C for 3 min, denatured at 98 °C for 15 s, cycled 8 times, annealed at 60 °C for 15 s, extended at 72 °C for 30 s, and held at 72 °C for 5 min to obtain fragments of approximately 300 bp size. Finally, it was sequenced using Illumina NovaSeq TM 6000 according to the sequencing mode PE150.

### 2.8. Methylation Level Detection

RNA samples were fragmented and immunoprecipitated magnetic beads were prepared, immunoprecipitated reaction was performed. Then, the obtained RNA samples were eluted and purified. The MeRIP RNA was analyzed together with the corresponding Input RNA by quantitative RT-PCR, or constructed using standard library building kits for RNA-seq library construction and identification of RNA methylation regions within the transcriptome by deep sequencing.

### 2.9. SiRNA Transfection

For the study, 1 × 10^6^ cells were inoculated into 6-well plates so that the cell density during transfection could reach 30~50%. After 24 h, samples were washed twice with DPBS, then 2 mL of the configured mixed reagent were added to each well. After 6 h, this was replaced with normal culture medium until the end of the corresponding transfection time when cells were collected for subsequent assay tests. To configure the mixed reagent, 10 µL of lipo2000 were added to 190 µL of serum-free culture medium, which was then gently mixed for 5 min. Next, 5 nmol of siRNA (sequence shown in Table 1) was added to 250 µL of DEPC water, i.e., dry powdered siRNA was configured into a 20 nmol/mL solution. Then, 5 µL of the prepared solution was combined with 195 µL of serum-free culture medium, and gently mixed for 5 min. The two liquid solutions described above were then mixed 1:1 and incubated for 15 min at room temperature to prepare the transfection complex. The complete culture medium was supplemented to 2 mL, which is the required transfection mixture for one well.

### 2.10. Cell Activity Assay

The CCK-8 colorimetric assay was used to determine cell activity [34]. Cells were inoculated on 96-well plates at an inoculation density of 1 × 10^4^/mL. After incubation with SP or BBR, 10 µL of CCK-8 solution was added to each well in a humidified incubator with 5% CO_2_ at 28 °C for 2 h. The absorbance value (OD) of each well was measured using ELISA (single wavelength, 450 nm).

### 2.11. Cellular TG Content Assay

Cells were inoculated into 96-well plates. The original cell culture medium was discarded after each group of cells was treated, followed by rinsing three times with DPBS. Next, 100 µL of cell lysis solution was added to each well and oscillated to fully lyse the cells. Intracellular TG level was measured with GPO-POD method using Triglyceride Colorimetric Assay Kit (Nanjing, Jiangsu, China). The content of protein in each sample was determined by BCA Protein Assay Kit (Nanjing, Jiangsu, China).

### 2.12. Total RNA Extraction, Reverse Transcription and Real-Time PCR

Total cellular RNA was extracted and reverse transcribed, and the expression level of *Camk1db* in each group of cells was detected by real-time fluorescence quantitative PCR. Real-time quantitative primer sequences for the genes to be tested were designed by Primer Premier 5.0 software and synthesized by Inweigeki (Table 2). The reaction solution was prepared on ice, mixed thoroughly, and the configured mixture was divided into reaction tubes of 18 µL each. Next, cDNA was added to the samples, which were then centrifuged thoroughly for the amplification reaction. The specific reaction conditions were as follows: 95 °C for 30 s; 95 °C for 5 s, 60 °C for 34 s, 40 cycles and 95 °C for 15 s; 60 °C for 60 s, 95 °C for 15 s. The relative expression of the genes to be tested in the samples was calculated using the 2^−ΔΔCT^ method [35].

### 2.13. Statistical Analysis

All results are shown as mean ± standard error (X ± SEM) and one-way ANOVA, and Duncan’s multiple range test were performed using SPSS 20.0. The significance level was *p* < 0.05, which was considered statistically significant.

## 3. Results

### 3.1. Modulation of Oxidative Stress, Apoptosis and Autophagy in Palmitic Acid-Induced Fat-Deposited Zebrafish Hepatocytes by Berberine

#### 3.1.1. Berberine on Sodium Palmitate-Induced Oxidative Stress in Zebrafish Hepatocytes

To detect the oxidative stress status of cells, the intracellular ROS (reactive oxygen species) content, MDA (malondialdehyde) content, and T-AOC (total antioxidant capacity), were measured. Excess ROS causes cellular damage, and MDA is an important marker of ROS-induced oxidative damage [36]. High levels of T-AOC can effectively protect cells from reactive oxygen species [37]. Four groups were created for the experiment: a control group (Control), a lipid accumulation group (SP), a berberine group (BBR), and a lipid accumulation + berberine group (SP + BBR), respectively. As seen in Figure 1A, the intracellular MDA content was significantly higher in the lipid accumulation group compared with the control group, but there was no significant change in the MDA content in the berberine group or the lipid accumulation + berberine group, and the MDA content in the lipid accumulation + berberine group was significantly reduced compared with the lipid accumulation group (*p* < 0.05). As seen in Figure 1B, the intracellular T-AOC (total antioxidant capacity) was significantly lower in the lipid accumulation group compared with the control group; there was no significant change in the T-AOC in the berberine group and the lipid accumulation + berberine group; and the T-AOC in the lipid accumulation + berberine group was significantly higher compared with the lipid accumulation group (*p* < 0.05). As seen in Figure 1C, the intracellular ROS (reactive oxygen species) content was significantly higher in the lipid accumulation group compared with the control group; there was no significant change in the berberine group; and the ROS content was significantly higher in the lipid accumulation + berberine group (*p* < 0.05). In addition, we also examined the ROS fluorescence graph, in which the brightness of the green spot represented the level of intracellular ROS content, and the brighter fluorescence indicated higher intracellular ROS concentration (red arrows). This can be seen from the luminescence trend compared with the control group, as the fluorescence intensity of all other groups was significantly higher, with the lipid accumulation + berberine group having the highest fluorescence intensity, which was consistent with the results in Figure 1D. These data indicated that lipid accumulation treatment caused oxidative stress in cells, and the total antioxidant capacity was reduced. Conversely, the addition of appropriate amounts of berberine were found to effectively alleviate the level of oxidative stress in cells, and improve the lipotoxicity caused by lipid accumulation. 

#### 3.1.2. Berberine on Sodium Palmitate-Induced Hepatocyte Apoptosis and Autophagy

To investigate changes of the apoptosis rates in each treatment group, flow cytometry was applied (Figure 2A), and the data were counted in Figure 2B plots (*p* < 0.05). Results indicated that the apoptosis rate in the lipid accumulation group was significantly higher than the control group, and the apoptosis rate in the berberine group was significantly lower, while the apoptosis rate in the lipid accumulation + berberine group was significantly lower than the lipid accumulation group. Among all groups, the apoptosis rate was lowest in the berberine group, followed by the control group and the lipid accumulation + berberine group, while the apoptosis rate was highest in the lipid accumulation group. To detect changes in the level of cellular autophagy in each group, Western blotting was applied to detect the expression of proteins associated with cellular autophagy in each group (Figure 2C). The relative content of P62 protein, which was calculated as shown in gray in Figure 2D, was significantly lower in the lipid accumulation group compared with the control group, significantly higher in the berberine group, and significantly higher in the lipid accumulation + berberine group, compared with the lipid accumulation group (*p* < 0.05). The relative content of LC3-B protein is presented in Figure 2E, indicating that LC3-B protein content was significantly higher in the lipid accumulation group compared with the control group, significantly lower in the berberine group, and significantly lower in the lipid accumulation + berberine group, compared with the lipid accumulation group (*p* < 0.05). The expression of P62 protein was negatively correlated with the level of cellular autophagy, and the expression of LC3-B protein was positively correlated with the level of cellular autophagy. The P62 and LC3-B protein expression levels indicated that the autophagy levels in the lipid accumulation group was significantly higher than in the control group, and that the autophagy level in the lipid accumulation + berberine group was significantly lower than in the lipid accumulation group, while the autophagy level in the berberine group was the lowest, and the autophagy level in the lipid accumulation group was the highest.

#### 3.1.3. Berberine on the Morphology of Zebrafish Hepatocytes Induced by Sodium Palmitate

As seen in Figure 1E, transmission electron micrographs of the cells were taken. The cells in the control group were morphologically intact, with round and regular shaped nuclei (red arrow), clear nuclear membranes, and normal endoplasmic reticulum and mitochondrial morphology. The morphology of the cells in the lipid accumulation group was significantly changed. The cells were severely vacuolated (green arrow); the nucleoli were extruded and deformed; the intracellular mitochondria were swollen (yellow arrow); and a large number of obvious lipid droplets appeared (blue arrow), indicating that the lipid accumulation treatment induced by sodium palmitate had an adverse effect on the cell morphology. In contrast, after the addition of an appropriate amount of berberine, there was a significant trend of cell morphology recovery compared with the lipid accumulation group, but there were still small amounts of swollen mitochondria and lipid droplets compared with the control group.

### 3.2. Effect of Berberine on Palmitic Acid-Induced m^6^A Methylation and Transcription Levels in Zebrafish Hepatocytes

#### 3.2.1. m^6^A Methylation Analysis

Cells were collected from four experimental groups: control group (Control), lipid accumulation group (SP), berberine group (BBR), and lipid accumulation + berberine group (SP + BBR), and methylation analysis was conducted by Merip-seq high-throughput sequencing to investigate the effect of berberine on lipid accumulation-induced m^6^A methylation in zebrafish cells. As seen in Table 3, in terms of the number of methylation sites, the highest number was detected in the berberine group, followed by the lipid accumulation + berberine group and the control group, with the least number of methylation sites detected in the lipid accumulation group. Results indicated that the addition of either lipid accumulation or berberine affected the number of m^6^A methylation sites in zebrafish hepatocytes. 

To study the distribution of m^6^A sites on mRNA, m^6^A peaks were then divided into four parts according to their position on the transcript: 5′ untranslated region (5′UTR); 3′ untranslated region (3′UTR), 1st Exon (1st Exon), and Other Exon (Other Exon). Appendix A and Figure 3 summarize the distribution of m^6^A sites in each group, with most peaks located in the 3′ untranslated region, followed by the 5′ untranslated region and other exon regions. Compared with the control group, there was a significant increase in the proportion of m^6^A peaks in the 3′ and 5′ untranslated regions in the lipid accumulation group, while the proportion within the 3′ untranslated region was significantly reduced and the proportion within 5′ untranslated region was further increased after the addition of berberine. M^6^A peaks possessed a higher enrichment ploidy. This indicated that lipid accumulation-induced zebrafish hepatocytes possessed different m^6^A methylation patterns, and the addition of appropriate amounts of berberine changed these m^6^A methylation patterns. 

The methylation and demethylation process of RNA requires binding proteins to the motifs of methylation sites. The analysis software HOMER was used to look for motifs with high feasibility in the peak region, as seen in Figure 4A, for the four groups of samples with motif prediction results. Motifs with multiple occurrences and high similarity in the four groups revealed potential motifs of zebrafish methylation sites. Results indicated that all four different treatment groups contained the “RRACH” motif, which indicated that zebrafish also conformed to the general pattern of other species, and confirmed that the results of this experiment were reliable and could be used for subsequent analysis. 

Appendix A illustrates the significant changes in methylation levels in each comparison group. The number of genes with significant methylation differences was 231 in the lipid accumulation group compared with the control group; 488 in the berberine group compared with the control group; 107 in the lipid accumulation + berberine group compared with the lipid accumulation group; and 441 in the lipid accumulation + berberine group compared with the berberine group. This indicated that the effect of the treatment with berberine on methylation levels was more significant compared to the lipid accumulation treatment. The distribution of the screened differentially methylated gene peaks was further counted, and data in Appendix A demonstrates that most of the differentially m^6^A methylated loci in each group was located in the 3′ untranslated region, followed by distribution in the 5′ untranslated region, then other exons, and finally, the first exon region. From the table, it can be seen that the proportion of m^6^A peaks in the 3′ and 5′ untranslated regions was significantly increased in the lipid accumulation versus control group, while the proportion of the 3′ untranslated region was significantly decreased and the proportion of the 5′ untranslated region was significantly increased in the berberine-repaired lipid accumulation group versus the control group. The addition of an appropriate amount of berberine was observed to change the m^6^A methylation pattern of differential genes in lipid accumulation-induced zebrafish hepatocytes, with a decrease in the proportion of the 3′ untranslated region and an increase in the proportion of the 5′ untranslated region. The differentially methylated peaks screened for each comparison group were subjected to motif analysis, and the top six ranked differentially methylated motifs of lipid accumulation versus control are listed in Figure 4B. From the figure, we can find that lipid accumulation treatment affected m^6^A methylation at these loci, and the m^6^A methylation occurring at these loci was also associated with berberine treatment. Similarly, these motifs contained multiple “RRACH” sequences, which verified the reliability of the results. 

#### 3.2.2. Transcriptome Analysis

Cells were collected for transcriptomic analysis from four experimental setup groups: control (Control), lipid accumulation group (SP), berberine group (BBR), and lipid accumulation + berberine group (SP + BBR). Significant changes in transcript levels in each comparison group are summarized in Appendix A. In terms of up- and down-regulation of transcript levels, the number of genes up-regulated by methylation was greater than the number of genes down-regulated by methylation in all comparison groups, except for the lipid accumulation + berberine group, where the number of genes up-regulated by methylation was smaller than the number of genes down-regulated by methylation compared with the lipid accumulation group.

The results of KEGG pathway enrichment analysis with significant differences in transcript levels are shown in Figure 5. Compared with the control group, the significantly differentially expressed genes in the lipid accumulation group were mainly involved in PPAR signaling pathway, phenylalanine, tyrosine and tryptophan biosynthesis, steroid biosynthesis, necroptosis, herpes simplex virus 1 infection, fructose and mannose metabolism, amino sugar and nucleotide sugar metabolism. Compared with the control group, the significantly differentially expressed genes in the berberine group were mainly involved in the TGF−beta signaling pathway, sphingolipid metabolism, selenocompound metabolism, primary bile acid biosynthesis, PPAR signaling pathway, FoxO signaling pathway, focal adhesion, arginine and proline metabolism, arachidonic acid metabolism, and the AGE−RAGE signaling pathway in diabetic complications. Compared with the control group, the significantly differentially expressed genes in the lipid accumulation + berberine group were mainly involved in the TGF−beta signaling pathway, terpenoid backbone biosynthesis, steroid biosynthesis, fatty acid elongation, cytokine−cytokine receptor interaction, biosynthesis of unsaturated fatty acids, and the AGE−RAGE signaling pathway in diabetic complications. Compared with the lipid accumulation + berberine group, the significantly differentially expressed genes in the lipid accumulation group were mainly involved in the TGF−beta signaling pathway, steroid biosynthesis, notch signaling pathway, MAPK signaling pathway, fatty acid elongation, biosynthesis of unsaturated fatty acids, and the AGE−RAGE signaling pathway in diabetic complications. Compared with the berberine group, the significantly differentially expressed genes in the lipid accumulation group were mainly involved in the sphingolipid metabolism, PPAR signaling pathway, FoxO signaling pathway, fatty acid degradation, cytokine−cytokine receptor interaction, biosynthesis of unsaturated fatty acids, and the AGE−RAGE signaling pathway in diabetic complications. Compared with the lipid accumulation + berberine group, the differentially expressed genes in the berberine group were mainly involved in the TGF−beta signaling pathway, steroid biosynthesis, taurine and hypotaurine metabolism, MAPK signaling pathway, linoleic acid metabolism, glycerolipid metabolism, cytokine−cytokine receptor interaction, and arachidonic acid metabolism. From the above results of the analysis of KEGG, the addition of high-fat and berberine significantly differentiates the transcriptional expression of genes in zebrafish liver cells, which in turn regulates related signaling pathways, mainly manifested as pathways related to lipid metabolism (PPAR signaling pathway, MAPK signaling pathway, glyceril lipid metabolism, etc.), and pathways related to oxidative stress (FoxO signaling pathway, TGF-beta signaling pathway, etc.). This indicates that lipid deposition and berberine treatment have obvious effects on lipid metabolism and oxidative stress in zebrafish hepatocytes.

#### 3.2.3. Analysis of m^6^A Methylation and Transcriptome Association

For the correlation analysis of transcription levels and methylation levels, we counted the number of genes with changes in methylation and transcription levels, respectively, and the number of genes with significant differences in methylation and transcription levels at the same time in each comparison group, as seen in Appendix A. We also analyzed the KEGG pathway for genes with significant differences in methylation and transcription together (Figure 6). Compared with the control group, the significantly differentially expressed genes in the lipid accumulation group were mainly involved in adrenergic signaling in cardiomyocytes. Compared with the control group, the significantly differentially expressed genes in the berberine group were mainly involved in PPAR signaling pathway, DNA replication, and arachidonic acid metabolism. Compared with the control group, the significantly differentially expressed genes in the lipid accumulation + berberine group were mainly involved in terpenoid backbone biosynthesis, synthesis and degradation of ketone bodies, ABC transporters, glycosaminoglycan biosynthesis − chondroitin sulfate/dermatan sulfate. Compared with the lipid accumulation + berberine group, the significantly differentially expressed genes in the lipid accumulation group were mainly involved in glycosaminoglycan biosynthesis − heparan sulfate/heparin, glycosaminoglycan biosynthesis − chondroitin sulfate/dermatan sulfate, beta−alanine metabolism, arginine biosynthesis, arginine and proline metabolism, and ABC transporters. Compared with the berberine group, the significantly differentially expressed genes in the lipid accumulation group were mainly involved in purine metabolism, PPAR signaling pathway, p53 signaling pathway, fatty acid degradation, arginine and proline metabolism, alanine, aspartate and glutamate metabolism, and N−Glycan biosynthesis. Compared with the lipid accumulation + berberine group, the differentially expressed genes in the berberine group were mainly involved in VEGF signaling pathway, tyrosine metabolism, sphingolipid metabolism, regulation of actin cytoskeleton, PPAR signaling pathway, MAPK signaling pathway, glycosaminoglycan biosynthesis − keratan sulfate, focal adhesion, adherens junction, and autophagy—other. From the above results of KEGG analysis of differentially methylated coding genes, it can be speculated that some genes related to lipid metabolism and translation regulation in zebrafish hepatocytes were differentially modified by m^6^A methylation under lipid accumulation and berberine treatment, and thus their expression was regulated, which in turn affected the phenotype.

#### 3.2.4. Screening of Target Genes

The screening process for methylation-transcription level association analysis focused on genes that showed significant changes in methylation levels and also showed significant differences in transcript levels (*p* < 0.05) (Fc ≥ 2). From these, the *Camk1db* gene of zebrafish was screened as the key gene for the next study. Trends are summarized in Appendix A. The M^6^A RNA methylation of the *Camk1db* gene was significantly lower in the lipid accumulation group compared with the control group, and the transcript level was reduced but not significantly different. The methylation levels and transcript levels were significantly higher in the berberine group compared to the lipid accumulation group. Methylation levels and transcript levels were also significantly increased in the lipid accumulation + berberine group compared to the lipid accumulation group.

### 3.3. Validation of Camk1db Gene Methylation and Its Regulation of Cellular Homeostasis in ZFL

#### 3.3.1. Validation of m^6^A Methylation Level and Transcription Level of Camk1db Gene

Changes in m^6^A methylation levels and gene expression levels occurring in the *Camk1db* gene in each treatment group were also evaluated at a low throughput level. Merip PCR was used to detect the m^6^A methylation level of the *Camk1db* gene (Figure 7A), and q-PCR was applied to detect the mRNA expression of the *Camk1db* gene (Figure 7B). Compared with the control group, the methylation and mRNA expression levels of *Camk1db* gene were significantly lower in the lipid accumulation group, and significantly higher in the berberine and lipid accumulation + berberine groups. Compared with the lipid accumulation group, the *Camk1db* gene methylation and mRNA expression levels were significantly higher in the lipid accumulation + berberine group. The trends in each set are basically consistent with the previous Merip-seq high-throughput sequencing results. 

#### 3.3.2. Regulation of Biological Functions of ZFL by Camk1db

To investigate the physiological function of *Camk1db* gene in zebrafish hepatocytes, siRNA transfection was used to silence *Camk1db* gene, and three groups were set up for the experiment: Control (control group), NC (non-specific control group), and Si-*Camk1db* (transfection group). First, we measured zebrafish physiological functions, including: cell activity, intracellular ROS content, apoptosis, and autophagy, before and after *Camk1db* gene silencing. As seen in Figure 7C, there was no significant change in cell activity in the NC group compared with the control group, while there was a significant decrease in cell activity after transfection with siRNA to silence the *Camk1db* gene, indicating that the expression of the *Camk1db* gene was related to cell activity. Combined with the analysis from the previous experimental results, the lipid accumulation treatment may have reduced the cell activity by inhibiting the expression of *Camk1db* gene. Berberine treatment may have enhanced the cell activity by promoting the expression of *Camk1db* gene. As seen in Figure 7D and Appendix A, no significant changes in intracellular ROS content were observed in the NC group compared with the control group, while there was a significant increase in intracellular ROS content after transfection with siRNA to silence the *Camk1db* gene, indicating that the expression of the *Camk1db* gene was negatively correlated with ROS. Combined with the analysis from the previous experimental results, lipid accumulation treatment promoted ROS by inhibiting the expression of the *Camk1db* gene. Berberine treatment increased the expression of *Camk1db* gene, which in turn inhibited the occurrence of ROS. As seen in Figure 7E and Appendix A, the apoptosis rate did not change significantly in the NC group compared with the control group, while there was a significant increase in the apoptosis rate after transfection with siRNA to silence the *Camk1db* gene, indicating that the expression of the *Camk1db* gene was negatively correlated with apoptosis. Combined with the analysis from the previous experimental results, lipid accumulation treatment promoted apoptosis by inhibiting the expression of the *Camk1db* gene. Berberine treatment increased the expression of *Camk1db* gene, which in turn inhibited the occurrence of apoptosis. To investigate changes in the level of autophagy occurring in the cells after transfection, the levels of P62 and LC3-B proteins in the cells before and after transfection were measured by Western blot. As seen in Figure 7F–H), there was a significant decrease in P62 protein content and a significant increase in LC3-B protein content in cells transfected with siRNA-silenced *Camk1db* gene compared with the control group. While P62 content was negatively correlated with the autophagy level of the cells, LC3-B content was positively correlated with the autophagy level of the cells. Therefore, the expression of *Camk1db* gene was inhibited and the level of cellular autophagy was increased. Combined with the analysis from the previous experimental results, these results indicated that lipid accumulation treatment inhibited the expression of *Camk1db* gene, which in turn increased the level of cellular autophagy. Berberine treatment increased the expression of *Camk1db* gene, which in turn inhibited the occurrence of autophagy in cells.

#### 3.3.3. Exploration of the Signaling Pathway Acting by Camk1db

The protein encoded by *Camk1db*, a calmodulin-dependent protein kinase, can be activated by Ca^2+^ and calmodulin binding in cells, which in turn activates certain intracellular signaling pathways, including the ERK (extracellular regulated protein kinases) pathway, by inducing cell depolarization. Michelle et al. reported that increased expression of CaMKII could activate the extracellular signal-regulated kinase ERK signaling pathway and increase the level of ERK phosphorylation [38]. The ERK signaling pathway is an important member of the cellular signaling pathway, which mediates different cellular functions, including cell proliferation, migration, differentiation, and survival. This is consistent with the function of *Camk1db* gene discussed above. Therefore, to further investigate whether the *Camk1db* gene affects downstream physiological indicators through the ERK pathway, Western blot was used to detect the levels of ERK and P-ERK protein in the cells before and after transfection, and the relative levels of P-ERK/ERK were further calculated to measure the activation level of the ERK pathway. Results are presented in Figure 7I,J. Compared with the control group, the relative levels of P-ERK/ERK did not change significantly in the NC group, while there was a significant decrease in the relative levels of P-ERK/ERK in cells transfected with siRNA to silence the Camk1db gene. To investigate whether lipid accumulation and berberine treatment could mediate the effect of *Camk1db* gene on the activation of the ERK pathway, the relative level of P-ERK/ERK in the four treatment groups (Control, SP, BBR, SP + BBR) identified above. Results are summarized in Figure 7K,L. Compared with the control group, the level of P-ERK/ERK was significantly lower in the lipid accumulation group and significantly higher in the berberine group, while the level of P-ERK/ERK was significantly higher in the lipid accumulation + berberine group compared with the lipid accumulation group. Taken together, these results suggested that lipid accumulation treatment can further inhibit the activation of the ERK pathway by suppressing the expression of *Camk1db* gene, and that berberine treatment could further promote the activation of ERK pathway by promoting the expression of *Camk1db* gene.

## 4. Discussion

M^6^A RNA methylation was important for dietary regulation of downstream genes and physiological indicators. In our study, berberine alleviated cellular oxidative stress by reducing the high levels of apoptosis and autophagy caused by lipid accumulations, and was firstly found to change the cellular m^6^A RNA methylation level. Berberine may increase the expression level of *Camk1db* mRNA by altering *Camk1db* m6A RNA methylation, and could regulate oxidative stress, apoptosis and autophagy through mediating the ERK1/2 signaling pathway activated by *Camk1db* in zebrafish-hepatocyte, which in turn affects cellular function, as shown in Figure 8. 

Under normal conditions, the oxidative and reductive systems in the cell are in organic equilibrium, maintaining normal cellular function. Once the oxidative and reductive systems are out of balance, homeostasis of the cellular redox environment ceases to exist, exhibiting stress and functional abnormalities, and continued imbalance will lead to abnormal cellular function. The free radical aging doctrine states that when oxidative damage dominates in cells, biomolecular functions will be disrupted, leading to the onset of aging [39]. It has been noted that excess palmitic acid can induce oxidative stress in hepatocytes [40]. In this study, berberine could effectively alleviate the oxidative stress, including ROS and MDA contents, of ZFL cells induced by lipid accumulation. Numerous studies have shown that ROS levels are closely associated with apoptosis and autophagy. Fat deposition leads to an increase in intracellular ROS content, which triggers endoplasmic reticulum stress and further causes apoptosis [41]. ROS can be involved in cell proliferation, differentiation, apoptosis and other physiological functions, especially mitochondria-derived ROS, which are closely involved in the regulation of cellular autophagy. Under oxidative stress, autophagy can, on the one hand, restore cells to normal function by removing damaged organelles, while conversely, it can also directly induce the death of oxidized cells [42]. In this experiment, we examined the changes of apoptosis and autophagy levels in each treatment group. The combined indices showed that the lipid accumulation treatment significantly elevated the apoptosis rate and autophagy level of cells compared with the control group, while the addition of berberine could alleviate the high level of apoptosis and autophagy induced by lipid accumulation. Therefore, the alleviating effect of berberine on the high levels of apoptosis and autophagy induced by lipid accumulation may be related to the oxidative stress status of the cells. Lipid accumulation treatment induces cellular oxidative stress and increases apoptosis and autophagy levels. Berberine may alleviate oxidative stress by reducing high levels of apoptosis and autophagy caused by lipid accumulation.

When cells are under different environments, m^6^A methylation can dynamically regulate the cellular response to the environment [43]. Several studies have shown that m^6^A methylation plays a crucial role in regulating oxidative stress and cell damage. For example, in the mouse renal tubular epithelial cell model treated with colistin [44] and the mouse model of hepatic ischemia-reperfusion [45], the oxidative stress response was altered with the down-regulation or overexpression of METTL3 or FTO. Mettl3-mediated modification of m^6^A is involved in PM2.5-induced apoptosis and autophagy [46]. METTL3 and FTO-mediated modification of m^6^A jointly participate in the oxidative stress process induced by cadmium sulfate and cause cell apoptosis [47]. This study was designed to estimate whether dietary factors or herb regulates cell proliferation by m^6^A methylation. We investigated the effect of berberine on lipid accumulation-induced m^6^A methylation in ZFL cells by Merip-seq high-throughput sequencing, and the results showed that m^6^A methylation was a dynamic and reversible modification occurring on RNA through changes in methylation sites and numbers, and that berberine treatment increased the overall m^6^A methylation level in zebrafish hepatocytes, while the methylation level of m^6^A decreased in cells under strong oxidative stress and cell damage after lipid accumulation treatment. This result is consistent with results reported by Sun [48]. Additionally, Andres et al. reported that the modification of m6A methylation in the 5′UTR increased during a state of oxidative stress, and that m6A methylation in the 5′UTR promoted the process of translation initiation of transcripts generated due to the stress state [49]. In this experiment, lipid accumulation treatment triggered the oxidative stress state of the cells. From the distribution observed, m6A methylation in the lipid accumulation group increased at the 5′UTR, and this proportion further increased after the addition of berberine, presumably related to the oxidative stress state. Berberine alleviated the oxidative stress state produced by lipid accumulation induced by increasing cellular m6A methylation modification in the 5′UTR. To investigate the gene function of the significantly altered m^6^A methylation, the differentially methylated genes KEGG pathway was enriched. The enrichment results showed that differentially methylated genes were involved in several pathways related to oxidative stress, including: TGF-beta signaling pathway, MAPK signaling pathway, FOXO signaling pathway, PPAR signaling pathway, and VEGF signaling pathway. It should be noted that the effect of berberine on lipid accumulation-induced m^6^A methylation production in ZFL cells was closely related to cellular oxidative stress, and m^6^A methylation modification regulated oxidative stress-related signaling pathways, thereby regulating the level of oxidative stress in cells and thus affecting cell proliferation phenotype.

To further investigate the mechanism of m^6^A methylation in the regulation of lipid accumulation-induced ZFL cells by berberine, a correlation analysis of methylation and transcript level data was performed, and results screened for differential genes with significant differential changes at both levels for *Camk1db*. The protein encoding the zebrafish *Camk1db* gene is CaMKID (calmodulin-dependent protein kinase ID), which is generally inactive, and its activation is closely related to Ca^2+^ and calmodulin activity within the cell. Ca^2+^, as a second messenger, is an important signaling ion that binds to receptors and proteins to participate in several physiological responses in the cell [50], particularly the protein CaM (calmodulin) [51]. When bound by Ca^2+^, CaM enters an activated state, allowing it to bind to the target protein, calmodulin-dependent protein kinase (CaMK), to exert physiological effects. CaMKI kinase is known to be widely present in the cytoplasm of many animal tissues and can promote cell proliferation by regulating cell cycle proteins [52]. The activated state of CaMKI can induce cell depolarization, which in turn activates the ERK (extracellular regulated protein kinases) pathway [53]. In this study, the effect of *Camk1db* gene silencing on the level of P-ERK/ERK was examined using siRNA silencing target gene combined with Western blotting. The results showed that the level of P-ERK/ERK was significantly reduced in the *Camk1db* gene silencing group, confirming the aforementioned findings that CaMKI can activate the ERK pathway. In addition, *Camk1db* gene silencing significantly reduced cell viability and elevated cellular oxidative stress, apoptosis and autophagy levels. It is hypothesized that the effect of *Camk1db* on the ERK pathway is closely related to the alteration of these physiological functions.

ERK is an extracellular signal-regulated protein kinase, which belongs to one of the MAPKs (mitogen-activated protein kinases) and is involved in several physiological processes such as cell growth, division, development, and death [54]. It has been shown that the ERK pathway is also involved in the process of oxidative stress, especially the reperfusion injury caused by hemorrhagic shock resuscitation is closely related to the ERK signaling pathway. The ERK signaling pathway is an important signaling pathway for post-ischemic drug treatment to play a protective role in tissue and organ ischemia-reperfusion injury. The activation of ERK can effectively improve ischemia-reperfusion injury [55]. ERK can promote survival by inhibiting the activation of pro-apoptotic BCL-2 family proteins (e.g., BAX and BIM) and thereby inducing the expression of anti-apoptotic members of the BCL-2 family (e.g., BCL-2, BCL-XL and MCL-1) [56]. In HT-29 cells, ERK can be inhibited by penicillin to up-regulate the expression of autophagy signature protein LC3 and promote the occurrence of autophagy [57]. In summary, phosphorylation of ERK1/2 activates the ERK pathway, promotes cell proliferation, inhibits oxidative stress and apoptosis, increases the autophagy level of cells, which is consistent with the results of this experiment. In addition, this study also found that the ERK phosphorylation level was reduced in the lipid accumulation group compared with the control group, and the ERK pathway was inhibited, while the treatment with berberine significantly increased the ERK phosphorylation level and activated the ERK pathway in ZFL cells. Combined with the effects of berberine on the methylation of lipid accumulation-induced ZFL cells and on the expression of *Camk1db* gene, it was hypothesized that berberine could regulate the expression level of mRNA by altering the M^6^A RNA methylation of the *Camk1db*, which further affected the synthesis of calmodulin-dependent protein kinase and thus the activation of the ERK pathway, resulting in changes in downstream physiological indicators.

## 5. Conclusions

In conclusion, this study explored the effects of berberine on the biological functions of oxidative stress, apoptosis, and autophagy in ZFL induced by sodium palmitate, while identified a new direction of m^6^A methylation of cellular RNA and screening the key differential gene *Camk1db*. However, there are still some unresolved questions. It will be necessary to elucidate the role of m^6^A methylation in the regulation of oxidative stress processes in vivo.

## Figures and Tables

**Figure 1 antioxidants-11-02370-f001:**
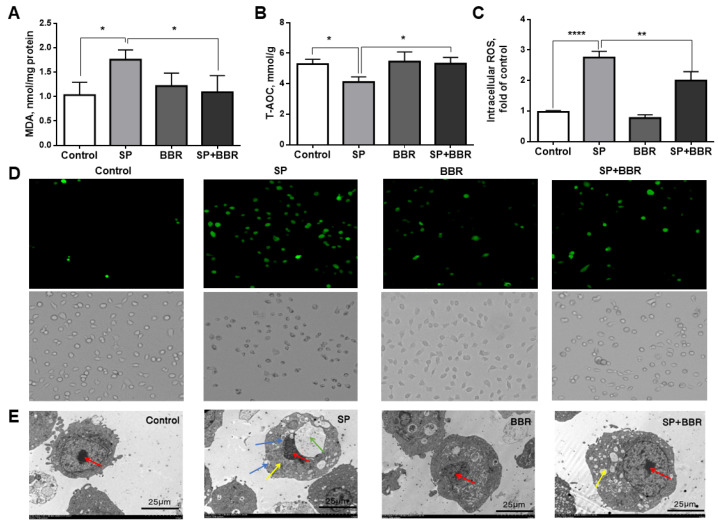
Indicators of oxidative stress in different treatment groups within ZFL and transmission electron microscopy pictures of ZFL. (**A**) is MDA (malondialdehyde). (**B**) is T-AOC (total antioxidant capacity). (**C**) is the relative expression of ROS. (**D**) is the ROS fluorescence microscope picture. The red arrow is the fluorescence of ROS in zebrafish liver cells. The higher the ROS content, the stronger the fluorescence intensity. Dates were presented as mean ± SEM (*n* = 3). * *p* < 0.05, ** *p* < 0.01, and **** *p* < 0.0001. (**E**) shows transmission electron microscopy pictures of ZFL. The red arrow is the nucleus, the blue arrow is the lipid drop, and the yellow arrow is the mitochondria, the green arrow is bubbles.

**Figure 2 antioxidants-11-02370-f002:**
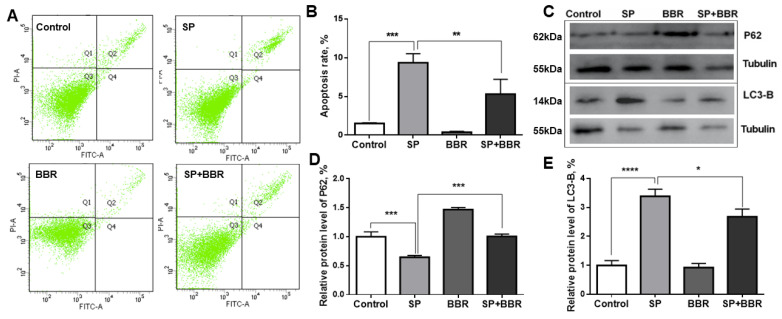
Berberine on sodium palmitate-induced hepatocyte apoptosis and autophagy. (**A**) is flow cytometry to detect ZFL apoptosis, each group has a cell count of 10,000. (**B**) is apoptosis and apoptosis rate of ZFL. (**C**) is WB diagram of P62 protein and LC3-B protein. (**D**) is relative expression level of P62 protein. (**E**) is relative expression level of LC3-B protein. Dates were presented as mean ± SEM (*n* = 3). * *p* < 0.05, ** *p* < 0.01, and *** *p* < 0.001, **** *p* < 0.0001.

**Figure 3 antioxidants-11-02370-f003:**
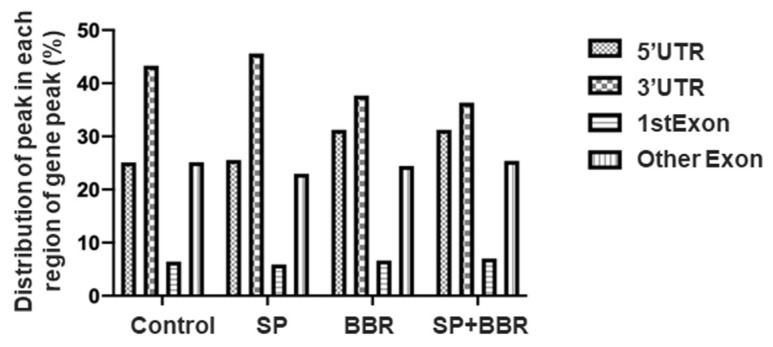
Distribution of m6A peaks of each group within ZFL.

**Figure 4 antioxidants-11-02370-f004:**
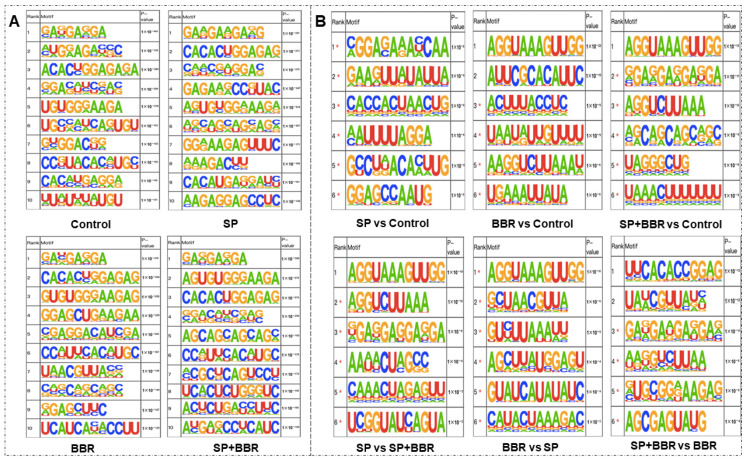
M^6^A methylation analysis. (**A**) is top ten motif analysis for different treatment groups within ZFL. (**B**) is top six motif analysis of each comparison group. *—possible false positive.

**Figure 5 antioxidants-11-02370-f005:**
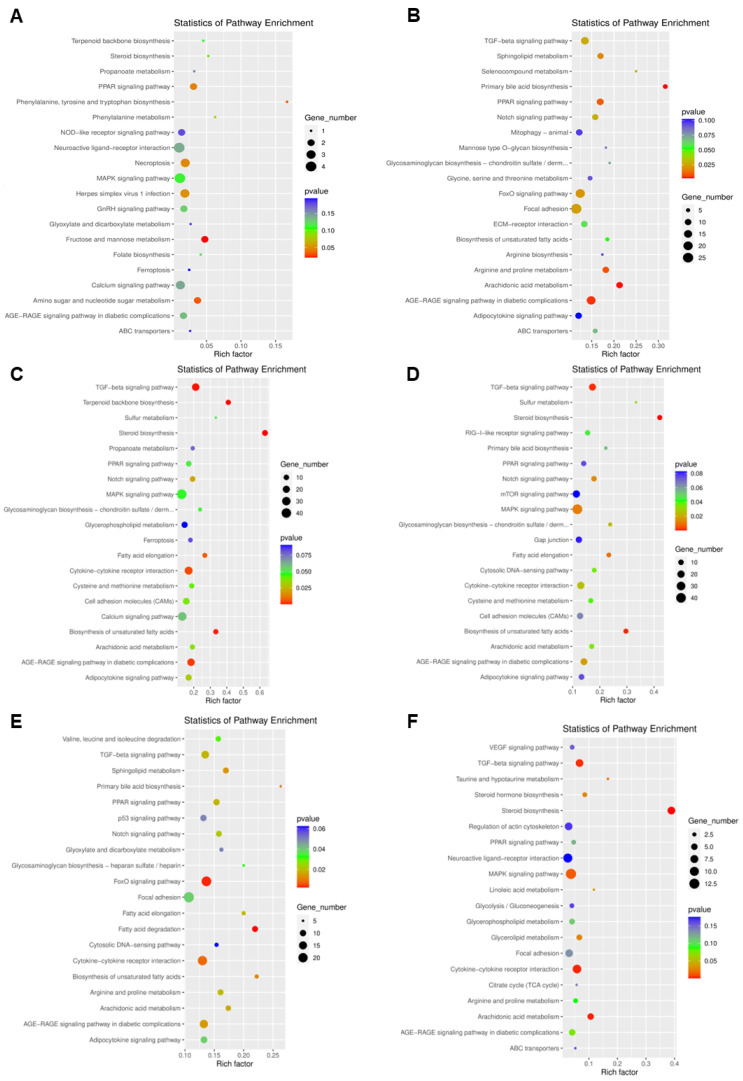
KEGG enrichment analysis of transcriptional differential genes in each comparison group. (**A**) is the comparison group of SP and Control. (**B**) is the comparison group of BBR and Control. (**C**) is the comparison group of SP + BBR and Control. (**D**) is the comparison group of SP + BBR and SP. (**E**) is the comparison group of BBR and SP. (**F**) is the comparison group of SP + BBR and BBR comparison group.

**Figure 6 antioxidants-11-02370-f006:**
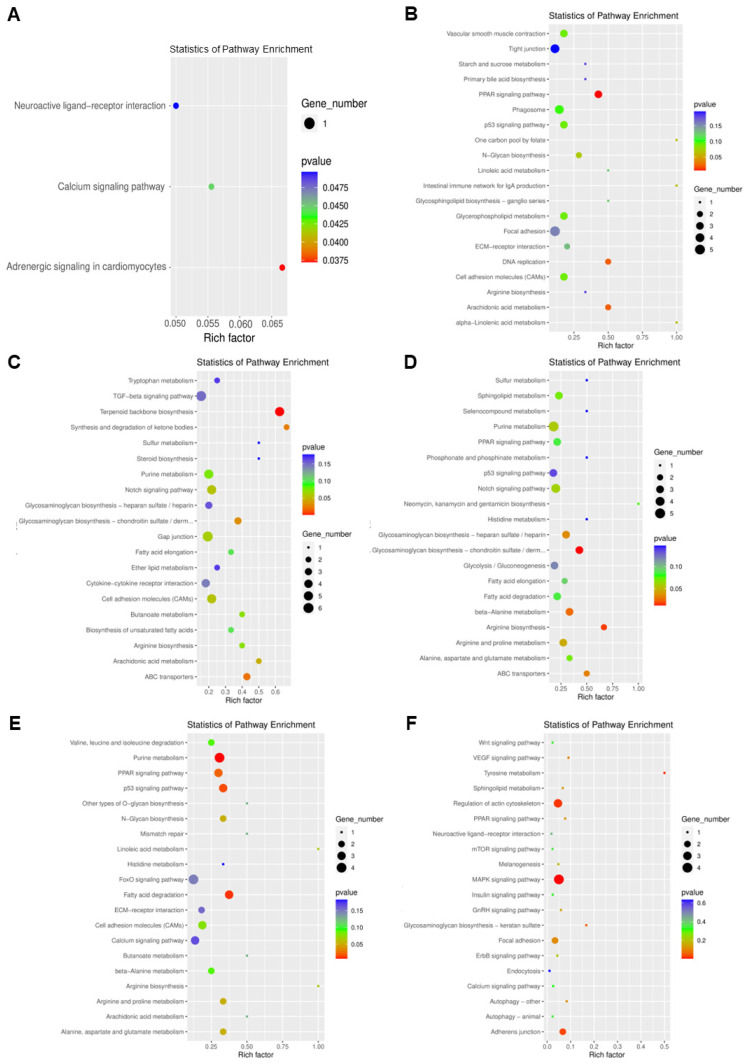
KEGG enrichment analysis of genes with significant differences in methylation and transcription levels in each comparison group. (**A**) is the comparison group of SP and Control. (**B**) is the comparison group of BBR and Control. (**C**) is the comparison group of SP + BBR and Control. (**D**) is the comparison group of SP + BBR and SP. (**E**) is the comparison group of BBR and SP. (**F**) is the comparison group of SP + BBR and BBR comparison group.

**Figure 7 antioxidants-11-02370-f007:**
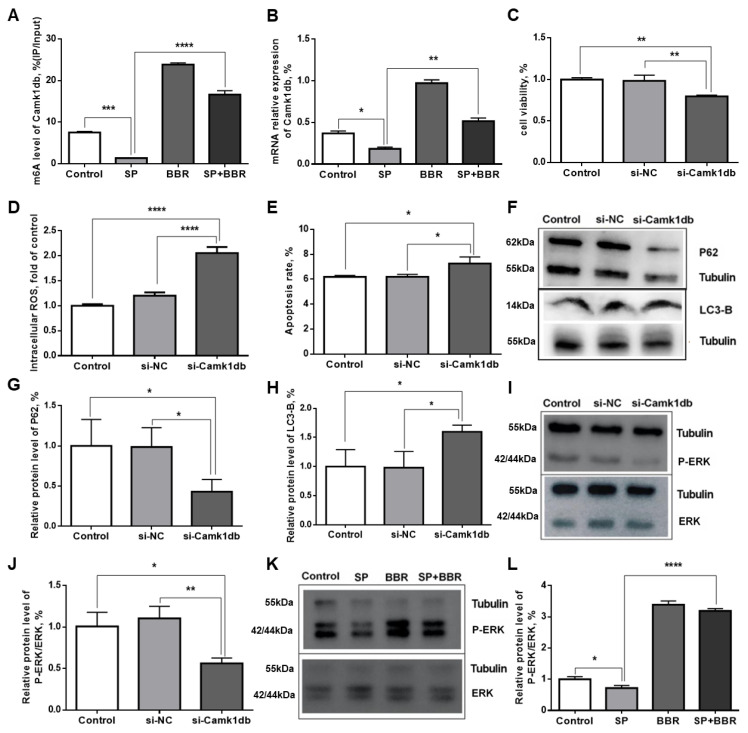
Validation of *Camk1db* gene methylation and its regulation of cellular homeostasis in ZFL by *Camk1db*. (**A**) is m^6^A methylation level of *Camk1db* gene in ZFL. (**B**) is relative mRNA expression of *Camk1db* gene in ZFL. (**C**) is liver cell activity. (**D**) is the relative expression of intracellular ROS. (**E**) is apoptosis rate. (**F**) is the WB map of LC3-B and P62 protein. (**G**) are relative content of P62 protein. (**H**) are relative content of LC3-B protein. (**I**,**J**) are the WB map and relative level of P-ERK/ERK protein before and after transfection. (**K**,**L**) are different treatment groups (Control, SP, BBR, SP + BBR) P-ERK/ERK WB map and relative expression of protein. Dates were presented as mean ± SEM (*n* = 3). * *p* < 0.05, ** *p* < 0.01, and *** *p* < 0.001, **** *p* < 0.0001.

**Figure 8 antioxidants-11-02370-f008:**
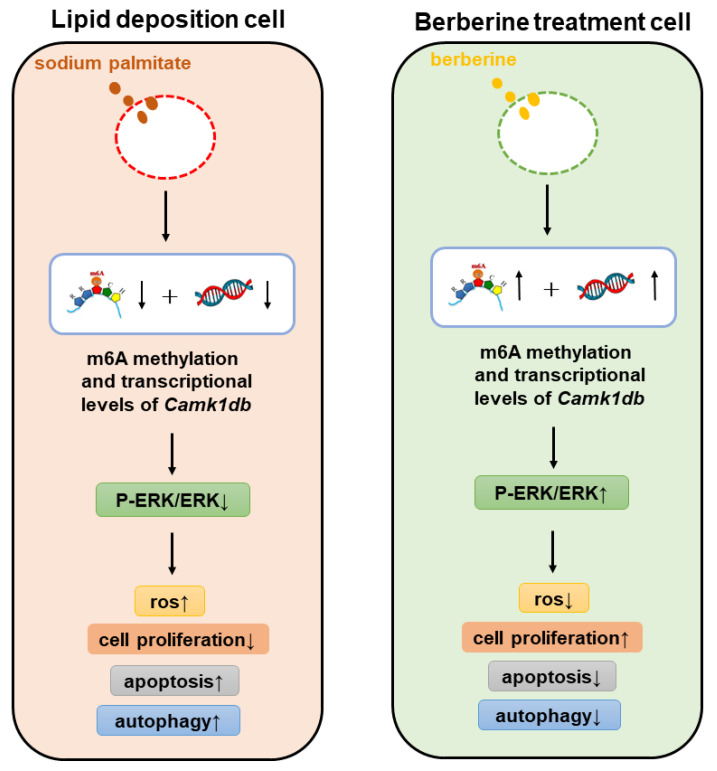
Berberine mediates the mechanism of *Camk1db*/ERK1/2 signaling pathway.

**Table 1 antioxidants-11-02370-t001:** SiRNA sequence of target genes.

Target Gene	Accession Number	siRNA Sequence
camk1db	NC_007136.7	CTGCAAGAACATCCACGAA

**Table 2 antioxidants-11-02370-t002:** Nucleotide sequences of primers of target genes.

Target Gene	Accession Number	Forward Primer (5′-3′)	Reverse Primer (5′-3′)
camk1db	NC_007136.7	GCGTGACGGATGGAGAAA	AGGCCACAGTAAACAGGAATAT

**Table 3 antioxidants-11-02370-t003:** Number of methylation sites.

Group	Number of Methylation Sites
Control	23,182
SP	20,859
BBR	26,152
SP + BBR	25,432

## Data Availability

All of the data is contained within the article and the Appendix A.

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
