# Peer review of "Berberine Regulation of Cellular Oxidative Stress, Apoptosis and Autophagy by Modulation of m^6^A mRNA Methylation through Targeting the *Camk1db*/ERK Pathway in Zebrafish-Hepatocytes"

_antioxidants, 2022, doi:10.3390/antiox11122370_

Round 1

Reviewer 1 Report

Authors reported detailed data on the effects of BRR, an ingredient of a Chinese medicine, on palmitic acid-induced toxicity in zebrafish-derived hepatocytes in culture. Studies are very important but some data seemed to be inconsistent without any comments and against Authors’ major proposal. Authors should also give additional explanation why they use zebrafish-derived hepatocytes in this study in Introduction. Be careful of spelling mistakes.

  • General points:

1) I don’t understand well the reason why Authors selected cultured hepatocyte cells of zebrafish to study the mechanism of BRR. Zebrafish has priority as in vivo model. If Authors used cultured cell system, they should study cultured hepatocyte derived from human and other mammals. Additional reason why ZFL cells were selected in this study is required. Many reports are found on BRR and mammalian hepatotoxicity in relation to methylation.

2) SP did not inhibit ROS formation (Figure 1C), although BRR blocked increased MDA content (1A) and MRR recovered reduction of total antioxidant capacity by SP. No effect of BRR on ROS fluorescence can be also confirmed in 1D. These inconsistent results on the effect of BRR on oxidative stress in Figure 1C and D is a serious problem against Authors’ major proposal in Figure 7.

3) Authors suggest that oxidative stress is a major cause of apoptosis and autophagy in ZFL cells. Which evidence do Authors have to support their proposal? Some data in Figure 1 could be inconsistent. Can Authors deny that oxidative stress and the other effects occurred in parallel fashion by SP?  Some papers reported involvement of CAMK in apoptosis (https://doi.org/10.3892/or.2020.7763), for example? How about the effect of antioxidant in Authors experiments, if they have?

Specific points:

(Abstract)

1)  A sentence in L18 and L19: I understood what Authors tried to say, but is this correct as English?

2)  A sentence in L48-L50: BBR not BRR?

3)  A sentence from L58 to L60 looks strange in the original article in the academic journal.

4) Figure 3B: Bar indication is difficult to see. Please modify it.

5) 3.2.2. Transcriptome analysis, second paragraph: Most important data should be Figure 5D (SPL vs SPL+BLL) in this section. But, description about this is almost lacking in the present form. Explanation on the other panels are also far from enough.

6) Figure 6: It seems that results of statistical analyses are very subtle. si-Camk1 seems not affect or rather increase slightly P62 band in F. Perhaps, Authors used tubulin band as control and vertical axis in G indicates calculated values of P62/Tubulin. Is this correct? If so, I don’t believe si-Cam1db reduced it compared to Control in panel G according to panel F. Can Authors replace panel F image with another one? Authors should explain the reason why they tried western blot with anti-tubulin antibody in Material and Methods. Please note that tublin should be replaced with tubulin in Figure 6.

Author Response

Response to Reviewer 1 Comments

Dear Editor:

Thank you for your letter and the comments concerning our manuscript. We had revised the manuscript according to the comments.

  • General points:

1)I don’t understand well the reason why Authors selected cultured hepatocyte cells of zebrafish to study the mechanism of BRR. Zebrafish has priority as in vivo model. If Authors used cultured cell system, they should study cultured hepatocyte derived from human and other mammals. Additional reason why ZFL cells were selected in this study is required. Many reports are found on BRR and mammalian hepatotoxicity in relation to methylation.

Answer #1: We apologize for the confusion caused by our incomplete “Introduction section”explanation in the article. Our study background is aquaculture industry. With the rapid development of scale and intensification in aquaculture, high-fat diets have been applied widely due to their lipoprotein-sparing effects, improved feed utilization, and reduced production costs. However, studies showed long-term excessive use of high-fat diet also has a series of negative effects , such as inducing metabolic disorders, producing anti-stress ability. The most significant effect is the occurrence of lipid deposition, resulting in lipid toxicity of fish and hepatic steatosis. Therefore, how to avoid the negative effects of high-fat diets in aquaculture industry is an important issue facing the aquaculture industry. Based on this, we also started a series of studied in vivo and vitro using fish [1-7]. Secondly, in our previous studies we found that berberine can effectively inhibit the deposition of fat in fish hepatopancreas induced by high-fat diet, attenuate oxidative stress, improve function of mitochondrial respiratory chain, reduce the apoptosis and inhibited inflammation response and modulate intestinal microflora profile caused by high-fat and high-carbohydrate diets. The aim of this study was to investigate whether berberine could regulate cellular oxidative stress, apoptosis and autophagy through changing the methylation level of cell gene m6A RNA. Zebrafish is the classic model organism for liver development and disease and there were many studies using Zebrafish hepatocytes to study fat deposition [8, 9]. In this study, the Zebrafish hepatocytes (CRL-2643) were purchased from the American Tissue Culture Collection (ATCC), which was a stable commercial cell line. So we used the ZFL cells. We have refined the above explanations in lines 64-66.

2)SP did not inhibit ROS formation (Figure 1C), although BRR blocked increased MDA content (1A) and MRR recovered reduction of total antioxidant capacity by SP. No effect of BRR on ROS fluorescence can be also confirmed in 1D. These inconsistent results on the effect of BRR on oxidative stress in Figure 1C and D is a serious problem against Authors’ major proposal in Figure 7.

Answer #2: The ROS level calculation formula[2] is in lines 116-117.

In Figures 1C, 1D, we re- checked and calculated ROS levels between groups. The intracellular ROS (reactive oxygen species) content was significantly higher in the lipid accumulation group compared with the control group; there was no significant change in the berberine group; and the ROS content was significantly higher in the lipid accumulation + berberine group (P<0.05). We had revised it.

3)Authors suggest that oxidative stress is a major cause of apoptosis and autophagy in ZFL cells. Which evidence do Authors have to support their proposal? Some data in Figure 1 could be inconsistent. Can Authors deny that oxidative stress and the other effects occurred in parallel fashion by SP?  Some papers reported involvement of CAMK in apoptosis (https://doi.org/10.3892/or.2020.7763), for example? How about the effect of antioxidant in Authors experiments, if they have?

Answer #3: Thank you for your question. In our article, we suggested that oxidative stress may be the main cause of apoptosis and autophagy in zebrafish cells, because a large number of studies have shown that oxidative stress levels are closely associated with apoptosis and autophagy. Oxidative stress induces apoptosis while promoting cellular autophagy [10]. Fat deposition leads to increased levels of intracellular oxidative stress, which further causes apoptosis [11]. In the state of oxidative stress, autophagy can, on the one hand, restore cells to normal function by removing damaged organelles; on the other hand, autophagy can also directly induce death of oxidized cells [12]. Therefore, the alleviating effect of berberine on the high level of apoptosis and autophagy induced by high lipid may be related to the oxidative stress state of cells. However, we do not deny that oxidative stress, apoptosis and autophagy may occur in parallel by high lipid induction, which all require us to further explore their relationship in subsequent experiments. Thanks again for pointing out the question, we have changed the wording in lines 600-603.

  • Specific points:

1)A sentence in L18 and L19: I understood what Authors tried to say, but is this correct as English?

Answer #1: Thanks for your careful observation of the correction, we have made the modifications in the first lines 15-16.

2)A sentence in L48-L50: BBR not BRR

Answer #2: Thank you for carefully pointing out the error for us. Berberine abbreviation is BBR, we have changed it in lines 47-49.

3)A sentence from L58 to L60 looks strange in the original article in the academic journal.

Answer #3: Thank you very much for the problem you pointed out. We have modified in lines 56-59.

4)Figure 3B: Bar indication is difficult to see. Please modify it.

Answer #4: Thank you for pointing out, we have adjusted the size of the histogram in Figure 3.

5)3.2.2. Transcriptome analysis, second paragraph: Most important data should be Figure 5D (SPL vs SPL+BLL) in this section. But, description about this is almost lacking in the present form. Explanation on the other panels are also far from enough.

Answer #5: Thank you very much for your advice. We have analyzed and explained the contents of this panel in more detail in lines 386-420.

6)Figure 6: It seems that results of statistical analyses are very subtle. si-Camk1 seems not affect or rather increase slightly P62 band in F. Perhaps, Authors used tubulin band as control and vertical axis in G indicates calculated values of P62/Tubulin. Is this correct? If so, I don’t believe si-Cam1db reduced it compared to Control in panel G according to panel F. Can Authors replace panel F image with another one? Authors should explain the reason why they tried western blot with anti-tubulin antibody in Material and Methods. Please note that tublin should be replaced with tubulin in Figure.

Answer #6: Thank you very much for pointing out the problem for us. For the calculation of the relative expression of P62 protein, we use the calculated value of P62/Tubulin. The ratio of the si-camk1db group is lower than that of the con group and the NC group. The results showed with western blot chart that can express our results more directly, see Figure 7F. Thank you again for pointing out the spelling mistake of Tubulin for us, we have corrected the marks in the Figure 2C, Figure 7F, 7I, 7K. We have also added an explanation of the choice of Tubulin as an intrinsic reference protein to the materials and methods in lines 78-83.

[1] C. Yu, M. Zhang, J. Liu, J. Zhang, J. Xu, W. Xu, Effects of sodium acetate on lipid metabolism, antioxidant capability and cell apoptosis of blunt snout bream (Megalobrama amblycephala) hepatocytes treated by sodium palmitate, Aquaculture Research, 53 (2022) 1098-1109.

[2] C. Yu, J. Zhang, Q. Qin, J. Liu, J. Xu, W. Xu, Berberine improved intestinal barrier function by modulating the intestinal microbiota in blunt snout bream (Megalobrama amblycephala) under dietary high-fat and high-carbohydrate stress, Fish Shellfish Immunol, 102 (2020) 336-349.

[3] S.S. Yang, C.B. Yu, Z. Luo, W.L. Luo, J. Zhang, J.X. Xu, W.N. Xu, Berberine attenuates sodium palmitate-induced lipid accumulation, oxidative stress and apoptosis in grass carp(Ctenopharyngodon idella)hepatocyte in vitro, Fish Shellfish Immunol, 88 (2019) 518-527.

[4] W.N. Xu, D.H. Chen, W.B. Liu, J.X. Xu, S.S. Yang, Molecular characterization of microtubule-associated protein 1-light chain 3B in Megalobrama amblycephala fed with high fat/berberine diets, Journal of applied genetics, 59 (2018) 345-355.

[5] W.N. Xu, D.H. Chen, Q.Q. Chen, W.B. Liu, Growth performance, innate immune responses and disease resistance of fingerling blunt snout bream, Megalobrama amblycephala adapted to different berberine-dietary feeding modes, Fish Shellfish Immunol, 68 (2017) 458-465.

[6] K.L. Lu, L.N. Wang, D.D. Zhang, W.B. Liu, W.N. Xu, Berberine attenuates oxidative stress and hepatocytes apoptosis via protecting mitochondria in blunt snout bream Megalobrama amblycephala fed high-fat diets, Fish Physiol Biochem, 43 (2017) 65-76.

[7] Q.Q. Chen, W.B. Liu, M. Zhou, Y.J. Dai, C. Xu, H.Y. Tian, W.N. Xu, Effects of berberine on the growth and immune performance in response to ammonia stress and high-fat dietary in blunt snout bream Megalobrama amblycephala, Fish Shellfish Immunol, 55 (2016) 165-172.

[8] Y.X. Pan, Z. Luo, M.Q. Zhuo, C.C. Wei, G.H. Chen, Y.F. Song, Oxidative stress and mitochondrial dysfunction mediated Cd-induced hepatic lipid accumulation in zebrafish Danio rerio, Aquatic toxicology (Amsterdam, Netherlands), 199 (2018) 12-20.

[9] M. Eide, M. Rusten, R. Male, K.H. Jensen, A. Goksøyr, A characterization of the ZFL cell line and primary hepatocytes as in vitro liver cell models for the zebrafish (Danio rerio), Aquatic toxicology (Amsterdam, Netherlands), 147 (2014) 7-17.

[10] V.O. Kaminskyy, B. Zhivotovsky, Free Radicals in Cross Talk Between Autophagy and Apoptosis, Antioxid Redox Signal, 21 (2013) 86-102.

[11] C. Ji, N. Kaplowitz, ER stress: can the liver cope?, J Hepatol, 45 (2006) 321-333.

[12] S. Sooparb, S.R. Price, J. Shaoguang, H.A. Franch, Suppression of chaperone-mediated autophagy in the renal cortex during acute diabetes mellitus, Kidney International, 65 (2004) 2135-2144.

Reviewer 2 Report

Dear authors,

your article titled: "Berberine regulation of cellular oxidative stress, apoptosis and autophagy by modulation of m6A mRNA methylation through targeting the Camk1db/ERK pathway in zebrafish-hepatocytes" shows some strenghts but also a lot of weaknesses.

Strenghts

1. The work is properly conducted and clearly demonstrates the

Weaknesses

1. why zebrafish liver cells should be a good model? Why don't you use also HepG2 or other mammals hepatic cells to compare effects of Berberine.

2. Why don't you use the entire zebrafish organism to observe also possible side effects of Berberine.

3. Berberine stimulates proliferation by activating phospho-ERK.....Does it stimulates cancer development?

4. Use of letters a and b in the graphs to indicate significance is not appropriate. It makes confusion. Use asterisks for significance and n.s for not significant results.

5. What happens to mitochondria after treatment with Berberine?

6. Use of qPCR to validate data from transcriptome analysis is not appropriate as qPCR is already performed during RNAseq . Protein checking by Western Blot is needed because of possible post-translational modifications.

7. KEGG pathways enrichment analysis is partly appropriated because not all genes involved in signalling pathways are considered (it is focused mostly on canonical signalling pathways while non-canonical pathways are less considered).

Kind regards

Author Response

Response to Reviewer 2 Comments

Dear Editor:

Thank you for your letter and the comments concerning our manuscript. We had revised the manuscript according to the comments.

1)Why zebrafish liver cells should be a good model? Why don't you use also HepG2 or other mammals hepatic cells to compare effects of Berberine.

Answer #1: We apologize for the confusion caused by our incomplete “Introduction section”explanation in the article. Our study background is aquaculture industry. With the rapid development of scale and intensification in aquaculture, high-fat diets have been applied widely due to their lipoprotein-sparing effects, improved feed utilization, and reduced production costs. However, studies showed long-term excessive use of high-fat diet also has a series of negative effects , such as inducing metabolic disorders, producing anti-stress ability. The most significant effect is the occurrence of lipid deposition, resulting in lipid toxicity of fish and hepatic steatosis. Therefore, how to avoid the negative effects of high-fat diets in aquaculture industry is an important issue facing the aquaculture industry. Based on this, we also started a series of studied in vivo and vitro using fish [1-7]. Secondly, in our previous studies we found that berberine can effectively inhibit the ectopic deposition of fat in fish hepatopancreas induced by high lipid intake, thus repairing the damaged lipid homeostasis and keeping the fish healthy in the early fish production trials. In this paper, we designed to investigate whether berberine could regulate cellular oxidative stress, apoptosis and autophagy by changing the methylation level of cell gene m6A RNA. Zebrafish is the classic model organism for liver development and disease and there were many studies using fish hepatocytes to study fat deposition [8-10]. In this study, the Zebrafish hepatocytes (CRL-2643) were purchased from the American Tissue Culture Collection (ATCC), which was a stable commercial cell line. We have refined the above explanations in lines 64-66.

2)Why don't you use the entire zebrafish organism to observe also possible side effects of Berberine.

Answer #2: Thank you for asking the question. We had studied the effect of berberine on growth performance, oxidative stress, lipid metabolism, cell apoptosis and intestinal barrier function in blunt snout bream under dietary high-fat and high-carbohydrate stress in vovo [2, 4-7]. In this paper, we designe to investigate the effect of berberine on m6A modification. So we used zebrafish liver cells for regulatory mechanism investigation in vitro.

3)Berberine stimulates proliferation by activating phospho-ERK.....Does it stimulates cancer development?

Answer #3: Thank you very much for your question. It was a good question. As we said above, berberine could attenuate oxidative stress, improve function of mitochondrial respiratory chain, reduce the apoptosis and inhibited inflammation response and modulate intestinal microflora profile caused by high-fat and high-carbohydrate diets. Based on in vivo experiments, we established cell model of berberine to repair high-fat damage, and screened out the optimal berberine concentration--25μM. The result showed berberine alleviated cellular oxidative stress through reducing the high levels of apoptosis and autophagy caused by lipid accumulations through mediating the ERK1/2 signaling pathway activated by Camk1db in zebrafish-hepatocyte. In this paper, we did not found cancer development.

4)Use of letters a and b in the graphs to indicate significance is not appropriate. It makes confusion. Use asterisks for significance and n.s for not significant results.

Answer #4: Thank you very much for your suggestions, we have modified the significant markers of difference in the figures 1,2,7.

5)What happens to mitochondria after treatment with Berberine?

Answer #5: Thank you for your question, which is very helpful for the future direction of our research. Some studies have now found that regulation of mitochondrial activity is also one of the ways in which berberine exerts its pharmacological effects. Phellodendron can regulate glucolipid metabolism by regulating mitochondrial activity, inhibiting hepatic gluconeogenesis, promoting glucagon-like peptide-1 (GLP-1) secretion, upregulating glycolysis, improving insulin sensitivity, and improving fatty acid oxidation. Xu et al. found that flavopiridol could inhibit mitochondrial respiratory chain complex I, thereby inhibiting ATP synthesis, enhancing glycolysis, and promoting glucose metabolism, a process independent of AMPK activation [11].Yao et al. explored the effects of flavopiridol on skeletal muscle lipid deposition and mitochondrial function through ex vivo experiments and found that flavopiridol significantly reduced triglyceride content in the gastrocnemius muscle of obese mice and It improved fatty acid oxidation, promoted mitochondrial biosynthesis and reduced abnormal ectopic lipid deposition in skeletal muscle through an AMPK/ pgc -1α-dependent manner [12]. All of the above studies suggest that berberine can target mitochondria to exert its pharmacological effects, and we may further analyze the mechanism of action of berberine on mitochondria function.

6)Use of qPCR to validate data from transcriptome analysis is not appropriate as qPCR is already performed during RNAseq . Protein checking by Western Blot is needed because of possible post-translational modifications.

Answer #6: Thank you very much for your question and we agree with your suggestion. Transcriptomics results can be verified by protein electrophoresis, but more and more studies have shown that we can also use Q-pcr to verify them [13, 14]. Secondly, because camk1db, this gene is currently relatively little studied on fish, and we cannot find a suitable primary antibody for protein electrophoresis, so we only use PCR for verification in this article.

7)KEGG pathways enrichment analysis is partly appropriated because not all genes involved in signalling pathways are considered (it is focused mostly on canonical signalling pathways while non-canonical pathways are less considered).

Answer #7: Thank you very much for your advice, we have performed a more detailed analysis of the enrichment of the KEGG pathway, analyzing all the signal paths that differ significantly in lines 386-420, 433-454.

[1] C. Yu, M. Zhang, J. Liu, J. Zhang, J. Xu, W. Xu, Effects of sodium acetate on lipid metabolism, antioxidant capability and cell apoptosis of blunt snout bream (Megalobrama amblycephala) hepatocytes treated by sodium palmitate, Aquaculture Research, 53 (2022) 1098-1109.

[2] C. Yu, J. Zhang, Q. Qin, J. Liu, J. Xu, W. Xu, Berberine improved intestinal barrier function by modulating the intestinal microbiota in blunt snout bream (Megalobrama amblycephala) under dietary high-fat and high-carbohydrate stress, Fish Shellfish Immunol, 102 (2020) 336-349.

[3] S.S. Yang, C.B. Yu, Z. Luo, W.L. Luo, J. Zhang, J.X. Xu, W.N. Xu, Berberine attenuates sodium palmitate-induced lipid accumulation, oxidative stress and apoptosis in grass carp(Ctenopharyngodon idella)hepatocyte in vitro, Fish Shellfish Immunol, 88 (2019) 518-527.

[4] W.N. Xu, D.H. Chen, W.B. Liu, J.X. Xu, S.S. Yang, Molecular characterization of microtubule-associated protein 1-light chain 3B in Megalobrama amblycephala fed with high fat/berberine diets, Journal of applied genetics, 59 (2018) 345-355.

[5] W.N. Xu, D.H. Chen, Q.Q. Chen, W.B. Liu, Growth performance, innate immune responses and disease resistance of fingerling blunt snout bream, Megalobrama amblycephala adapted to different berberine-dietary feeding modes, Fish Shellfish Immunol, 68 (2017) 458-465.

[6] K.L. Lu, L.N. Wang, D.D. Zhang, W.B. Liu, W.N. Xu, Berberine attenuates oxidative stress and hepatocytes apoptosis via protecting mitochondria in blunt snout bream Megalobrama amblycephala fed high-fat diets, Fish Physiol Biochem, 43 (2017) 65-76.

[7] Q.Q. Chen, W.B. Liu, M. Zhou, Y.J. Dai, C. Xu, H.Y. Tian, W.N. Xu, Effects of berberine on the growth and immune performance in response to ammonia stress and high-fat dietary in blunt snout bream Megalobrama amblycephala, Fish Shellfish Immunol, 55 (2016) 165-172.

[8] D. Qianwen, Z. Zhen, R. Chao, D. Zhenyu, Z. Zhigang, Establishment of a model for oleic acid-induced lipid droplet formation and degradation in zebrafish hepatocytes, Journal of Agricultural Science and Technology, 20 (2018) 129-138.

[9] Y.X. Pan, Z. Luo, M.Q. Zhuo, C.C. Wei, G.H. Chen, Y.F. Song, Oxidative stress and mitochondrial dysfunction mediated Cd-induced hepatic lipid accumulation in zebrafish Danio rerio, Aquatic toxicology (Amsterdam, Netherlands), 199 (2018) 12-20.

[10] M. Eide, M. Rusten, R. Male, K.H. Jensen, A. Goksøyr, A characterization of the ZFL cell line and primary hepatocytes as in vitro liver cell models for the zebrafish (Danio rerio), Aquatic toxicology (Amsterdam, Netherlands), 147 (2014) 7-17.

[11] M. Xu, Y. Xiao, J. Yin, W. Hou, X. Yu, L. Shen, F. Liu, L. Wei, W. Jia, Berberine promotes glucose consumption independently of AMP-activated protein kinase activation, PLoS One, 9 (2014) e103702.

[12] S. Yao, Y. Yuan, H. Zhang, X. Meng, L. Jin, J. Yang, W. Wang, G. Ning, Y. Zhang, Z. Zhang, Berberine attenuates the abnormal ectopic lipid deposition in skeletal muscle, Free radical biology & medicine, 159 (2020) 66-75.

[13] C. Liu, Y. Liu, L. Liang, S. Cui, Y. Zhang, RNA-Seq based transcriptome analysis during bovine viral diarrhoea virus (BVDV) infection, BMC genomics, 20 (2019) 774.

[14] M. Fu, H. Su, Z. Su, Z. Yin, J. Jin, L. Wang, Q. Zhang, X. Xu, Transcriptome analysis of Corynebacterium pseudotuberculosis-infected spleen of dairy goats, Microbial pathogenesis, 147 (2020) 104370.

Reviewer 3 Report

This paper is generally well written and pretty clear.  It represents a large body of work and deserves to be published.  I have some minor request for revisions, which can be addressed with some clarifications in the text.

I assume this is to investigate mechanisms related to inflammation associated with diet and cancer, so the rationale for using zebrafish hepatocytes really is not clear.

In Figure one the different color arrows are not evident.  I can’t tell a “bubble” from a lipid droplet and can’t seea green arrow at all.

Figure 3 emphasizes the differences in M6A RNA methylation motifs, but the main result discussed emphasizes location.  If I am understanding correctly, palmitate treatment increases 3’ m6A and berberine shifts the 3’ UTR methylation to 5’ UTR, presumably altering stability or translation.  Figure 3B makes the point and is the hardest to read.  I would ask that it be larger and easier to read.  It seems that the supplemental table 3 is more relevant to the story than the sequences shown in 3A and C, which I can’t read anyway!  Maybe make those supplemental and the actual m6A numbers part of Figure 3.

I could not read the KEGG figures at all.  Please consider changing the font.

On the bottom of page 11 (lines 425- 427) the authors write that methylation of the camk1db gene (I assume this is m6A methylation of the RNA) was significantly lowered, but transcript levels were no significantly lowered.  What then was the rationale for pursuing camk1db as a target gene?

Later in the validation studies the levels were both significantly lower “like the initial transcriptomic and methylation screen”.  Please resolve this inconsistency.

P12 lines 449-451 states “First we measured …before and after intracellular ROS gene silencing.”  Do you mean siRNA gene silencing?

Related to this the term “hepatocyte activity” (line450) and cell viability in Figure 6C seem to indicate the same thing.  What does this mean?  Explain with a sentence.

In Figure 6 there’s “relative protein level of P-ERK/ERK %” What does this mean? Was the level of ERK really reduced, as indicated on line 610, or was phosphorylated ERK reduced?  Because it looks like phosphorylation is what changes.

The first paragraph of the discussion (lines 527-531) says “Berberine increased expression level of Camk1db mRNA by altering the methylation level of the Camk1db gene…”.   Again, I assume this means m6A RNA methylation, not methylation of the gene (i.e. promoter regions or CpG islands), which is not discussed in this paper.  How do then propose that transcript level was increased by m6A methylation?

Discussion line 582-583, says “m6A methylation in the lipid accumulation group increased at the 5’ end, and this proportion increased with berberine treatment…” but Figure 3B doesn’t show this.  Lipid treatment shows an increase in the 3’ UTR, which appears to shift to 5’ with berberine treatment.  Please clarify what you mean here.

Author Response

Response to Reviewer 1 Comments

Dear Editor:

Thank you for your letter and the comments concerning our manuscript. We had revised the manuscript according to the comments.

1)I assume this is to investigate mechanisms related to inflammation associated with diet and cancer, so the rationale for using zebrafish hepatocytes really is not clear.

Answer #1: We apologize for the confusion caused by our incomplete explanation in the article. The zebrafish (Danio rerio) is a widely used model species in biomedical research. The ZFL cell line, established from zebrafish liver, and freshly isolated primary hepatocytes from zebrafish have been used in several toxicological and mechanism studies. And our study background is aquaculture industry. With the rapid development of scale and intensification in aquaculture, high-fat diets have been applied widely due to their lipoprotein-sparing effects, improved feed utilization, and reduced production costs. However, studies showed long-term excessive use of high-fat diet also has a series of negative effects , such as inducing metabolic disorders, producing anti-stress ability. The most significant effect is the occurrence of lipid deposition, resulting in lipid toxicity of fish and hepatic steatosis. Therefore, how to avoid the negative effects of high-fat diets in aquaculture industry is an important issue facing the aquaculture industry. Based on this, we also started a series of studied in vivo and vitro using fish [1-7]. Secondly, in our previous studies we found that berberine can effectively inhibit the deposition of fat in fish hepatopancreas induced by high-fat diet, attenuate oxidative stress, improve function of mitochondrial respiratory chain, reduce the apoptosis and inhibited inflammation response and modulate intestinal microflora profile caused by high-fat and high-carbohydrate diets. The aim of this study was to investigate whether berberine could regulate cellular oxidative stress, apoptosis and autophagy through changing the methylation level of cell gene m6A RNA. Zebrafish is the classic model organism for liver development and disease and there were many studies using fish hepatocytes to study fat deposition [8-10]. In this study, the Zebrafish hepatocytes (CRL-2643) were purchased from the American Tissue Culture Collection (ATCC), which was a stable commercial cell line. We have refined the above explanations in lines 64-66.

2)In Figure one the different color arrows are not evident.  I can’t tell a “bubble” from a lipid droplet and can’t seea green arrow at all.

Answer #2: Thanks for the question, which has been re-annotated in Figure 1.

3)Figure 3 emphasizes the differences in m6A RNA methylation motifs, but the main result discussed emphasizes location.  If I am understanding correctly, palmitate treatment increases 3’ m6A and berberine shifts the 3’ UTR methylation to 5’ UTR, presumably altering stability or translation.  Figure 3B makes the point and is the hardest to read.  I would ask that it be larger and easier to read.  It seems that the supplemental table 3 is more relevant to the story than the sequences shown in 3A and C, which I can’t read anyway!  Maybe make those supplemental and the actual m6A numbers part of Figure 3.

Answer #3: Thank you for your suggestion. In this paragraph, we have mainly analyzed two parts. The first part is the analysis of m6A RNA methylation in four different groups, we first performed a site analysis in Table 3 and S1, Figure 3, where the results show that the palmitate treatment increased by 3' UTR, and berberine converted 3' UTR methylation to 5' UTR, which may change stability or translation; The analysis of m6A RNA methylation motifs is shown in Figure 4A, where the results show that zebrafish also conform to the general pattern of other species, and that four different treatment groups contain motifs of "RRACH", indicating that zebrafish also conform to the general rules of other species, resulting in the reliability of the results of this experiment that can be used for subsequent analysis. The second part is the m6A RNA methylation analysis after comparison between two of four different groups, the purpose is to analyze the differential methylation genes, Table S2 lists the changes in the number of significant genes in the methylation level in each comparison group, and then the distribution of the filtered differential methylation gene peaks is statistical, the methylation sites of the changes are shown in Table S3, and then the differential methylation peaks screened by each comparison group are motif analysis. The possible motifs of differential methylation sites were selected for two different treatment groups, and similar conclusions were obtained to the above motifs containing multiple sequences of "RRACH", which verified the reliability of the results. In order to make the article more logically coherent, we have refined the article in lines 325-337, adjusted the data position to change the supplementary table 1 to table 3, and changed the size of Figure 3B for clearer observation.

4)I could not read the KEGG figures at all.  Please consider changing the font.

Answer #4: Thank you for your suggestion. We have changed the font and graphic size as shown in Figures 5-6.

5)On the bottom of page 11 (lines 425- 427) the authors write that methylation of the camk1db gene (I assume this is m6A RNA methylation of the RNA) was significantly lowered, but transcript levels were no significantly lowered.  What then was the rationale for pursuing camk1db as a target gene?

Answer #5: Thank you very much for the problem you pointed out. To explore the mechanism of action of berberine, we first performed transcriptomic analysis on four different treatment groups, and because more and more studies have shown that m6A RNA methylation plays an important role in regulating lipid deposition, because we also performed Merip-seq of methylation levels in four different treatment groups to explore whether berberine functions through methylation levels within zebrafish cells, we screened the camk1db gene. We found that the mRNA expression in the camk1db gene was significantly higher in the high-fat plus berberine group compared with the high-fat group. Compared with the control group, mRNA expression in the high-fat group was reduced, although the difference was not significant. In addition, we also unexpectedly found that the methylation level of camk1db gene was significantly higher than that in the high-fat group, and the mRNA expression in the high-fat plus berberine group was significantly increased. We were surprised that mRNA expression in the high-fat group was significantly lower compared to the control group, so we wanted to explore whether berberine plays a role by regulating the methylation level of Camk1db. However, due to the false positives of high-throughput sequencing, in order to ensure the accuracy of the experiment, we used the immunomagnetic bead co-precipitation method to detect the methylation level expression of genes between groups at low throughput levels, and Q-pcr to detect the expression of gene mRNA between groups, and found that the methylation level and mRNA expression of Camk1db gene in the high-fat plus berberine group were significantly increased compared with the high-fat group. Compared with the control group, the methylation level and mRNA expression of Camk1db gene in the high-fat group were significantly reduced, and the results are shown in Fig. 7A, 7B, and finally camk1db was selected as our target gene for study.

6)Later in the validation studies the levels were both significantly lower “like the initial transcriptomic and methylation screen”.  Please resolve this inconsistency.

Answer #6: Thank you very much teacher for the problem you pointed out. I'm sorry that our wording of this passage is not rigorous enough. Since RNA-seq is used for large-scale screening, reflecting the overall gene expression trend of the sample, there may be some inconsistencies in the change trend of each gene. To avoid this, we verified the results at a low-throughput level. We compared the methylation level and transcription level of the four treatment groups according to the high-throughput grouping. Among them, the trend of upward and low-level changes in 12 results was completely consistent with the high-throughput sequencing results, and the significant trend in 11 results was consistent with the high-throughput sequencing results, so here we concluded that the trend of each group was basically consistent with the Merip-seq high-throughput sequencing results. We have made more rigorous changes to the wording in lines 488-490.

7) P12 lines 449-451 states “First we measured …before and after intracellular ROS gene silencing.”  Do you mean siRNA gene silencing?

Answer #7: Sorry for making this mistake due to our carelessness. Here we are trying to express the camk1db gene silencing. We have made changes in the text in lines 493-495.

8)Related to this the term “hepatocyte activity” (line450) and cell viability in Figure 6C seem to indicate the same thing.  What does this mean?  Explain with a sentence.

Answer #8: Thank you for your careful comments. The “hepatocyte activity”and the “cell viability” in Figure mean the same thing. For consistency in terminology, we have corrected it in the text, see line 494.

9)In Figure 6 there’s “relative protein level of P-ERK/ERK %” What does this mean? Was the level of ERK really reduced, as indicated on line 610, or was phosphorylated ERK reduced?  Because it looks like phosphorylation is what changes.

Answer #9: Thank you for your question. We had revised “relative protein level of P-ERK/ERK %” to “the level of p-ERK/ERK%”, which means the phosphorylation level of ERK [11, 12]. We have corrected see lines 653-655.

10)The first paragraph of the discussion (lines 527-531) says “Berberine increased expression level of Camk1db mRNA by altering the methylation level of the Camk1db gene…”.   Again, I assume this means m6A RNA methylation, not methylation of the gene (i.e. promoter regions or CpG islands), which is not discussed in this paper.  How do then propose that transcript level was increased by m6A RNA methylation?

Answer #10: Thank you for your question. The first paragraph of the discussion says that "Berberine increases the expression level of Camk1db mRNA by altering the methylation level of the Camk1db gene" refers to m6A RNA methylation of the Camk1db, not DNA methylation. We had revised it, now line 571. The m6A RNA methylation modification was closely related to the expression of genes, it can promote the start of mRNA translation process [13-15]. The mRNA expression level of Cank1db gene in the berberine/high-fat group was significantly increased compared to the high-fat group. So we estimated whether berberine may regulate the gene mRNA expression by m6A RNA methylation.

11)Discussion line 582-583, says “m6A RNA methylation in the lipid accumulation group increased at the 5’ end, and this proportion increased with berberine treatment…” but Figure 3B doesn’t show this.  Lipid treatment shows an increase in the 3’ UTR, which appears to shift to 5’ with berberine treatment.  Please clarify what you mean here.

Answer #11: Thank you very much for your questions. The analysis and statistical results of methylation sites are shown in Table S1, and the bar chart is drawn in Figure 3. Combined with the chart, we can find that the m6A RNA methylation site in the high fat group increased by 0.43% at the 5’UTR compared with the control group, while the m6A RNA methylation site in the high fat group increased by 5.67% at the 5’UTR compared with the high fat group. Therefore, we discussed that " m6A RNA methylation in the lipid accumulation group increased at the 5’UTR, and this proportion increased with berberine treatment ". Of course, in addition, the high fat group compared with the control group m6A RNA methylation site at the 3’UTR increased by 2.3%, and the high fat group compared with the high fat group m6A RNA methylation site at the 3’UTR decreased by 9,24%. Therefore, we analyzed in lines 325-337 that compared with the control group, the proportion of m6A peaks in the 3’UTR and 5’UTR increased in the high-fat group, while after the addition of berberine, the proportion of 3’UTR decreased, and the proportion of 5’UTR increased further, and m6A peaks had a higher enrichment ratio. These results indicated that high-fat induced zebrafish liver cells had different m6A RNA methylation patterns, and the addition of appropriate amount of berberine changed the m6A RNA methylation patterns.

[1] C. Yu, M. Zhang, J. Liu, J. Zhang, J. Xu, W. Xu, Effects of sodium acetate on lipid metabolism, antioxidant capability and cell apoptosis of blunt snout bream (Megalobrama amblycephala) hepatocytes treated by sodium palmitate, Aquaculture Research, 53 (2022) 1098-1109.

[2] C. Yu, J. Zhang, Q. Qin, J. Liu, J. Xu, W. Xu, Berberine improved intestinal barrier function by modulating the intestinal microbiota in blunt snout bream (Megalobrama amblycephala) under dietary high-fat and high-carbohydrate stress, Fish Shellfish Immunol, 102 (2020) 336-349.

[3] S.S. Yang, C.B. Yu, Z. Luo, W.L. Luo, J. Zhang, J.X. Xu, W.N. Xu, Berberine attenuates sodium palmitate-induced lipid accumulation, oxidative stress and apoptosis in grass carp(Ctenopharyngodon idella)hepatocyte in vitro, Fish Shellfish Immunol, 88 (2019) 518-527.

[4] W.N. Xu, D.H. Chen, W.B. Liu, J.X. Xu, S.S. Yang, Molecular characterization of microtubule-associated protein 1-light chain 3B in Megalobrama amblycephala fed with high fat/berberine diets, Journal of applied genetics, 59 (2018) 345-355.

[5] W.N. Xu, D.H. Chen, Q.Q. Chen, W.B. Liu, Growth performance, innate immune responses and disease resistance of fingerling blunt snout bream, Megalobrama amblycephala adapted to different berberine-dietary feeding modes, Fish Shellfish Immunol, 68 (2017) 458-465.

[6] K.L. Lu, L.N. Wang, D.D. Zhang, W.B. Liu, W.N. Xu, Berberine attenuates oxidative stress and hepatocytes apoptosis via protecting mitochondria in blunt snout bream Megalobrama amblycephala fed high-fat diets, Fish Physiol Biochem, 43 (2017) 65-76.

[7] Q.Q. Chen, W.B. Liu, M. Zhou, Y.J. Dai, C. Xu, H.Y. Tian, W.N. Xu, Effects of berberine on the growth and immune performance in response to ammonia stress and high-fat dietary in blunt snout bream Megalobrama amblycephala, Fish Shellfish Immunol, 55 (2016) 165-172.

[8] D. Qianwen, Z. Zhen, R. Chao, D. Zhenyu, Z. Zhigang, Establishment of a model for oleic acid-induced lipid droplet formation and degradation in zebrafish hepatocytes, Journal of Agricultural Science and Technology, 20 (2018) 129-138.

[9] Y.X. Pan, Z. Luo, M.Q. Zhuo, C.C. Wei, G.H. Chen, Y.F. Song, Oxidative stress and mitochondrial dysfunction mediated Cd-induced hepatic lipid accumulation in zebrafish Danio rerio, Aquatic toxicology (Amsterdam, Netherlands), 199 (2018) 12-20.

[10] M. Eide, M. Rusten, R. Male, K.H. Jensen, A. Goksøyr, A characterization of the ZFL cell line and primary hepatocytes as in vitro liver cell models for the zebrafish (Danio rerio), Aquatic toxicology (Amsterdam, Netherlands), 147 (2014) 7-17.

[11] F. Liu, X.X. Feng, S.L. Zhu, H.Y. Huang, Y.D. Chen, Y.F. Pan, R.R. June, S.G. Zheng, J.L. Huang, Sonic Hedgehog Signaling Pathway Mediates Proliferation and Migration of Fibroblast-Like Synoviocytes in Rheumatoid Arthritis via MAPK/ERK Signaling Pathway, Frontiers in immunology, 9 (2018) 2847.

[12] Z. Jia, J. Yang, Z. Cao, J. Zhao, J. Zhang, Y. Lu, L. Chu, S. Zhang, Y. Chen, L. Pei, Baicalin ameliorates chronic unpredictable mild stress-induced depression through the BDNF/ERK/CREB signaling pathway, Behavioural brain research, 414 (2021) 113463.

[13] X. Wang, B.S. Zhao, I.A. Roundtree, Z. Lu, D. Han, H. Ma, X. Weng, K. Chen, H. Shi, C. He, N(6)-methyladenosine Modulates Messenger RNA Translation Efficiency, Cell, 161 (2015) 1388-1399.

[14] T. Liu, Q. Wei, J. Jin, Q. Luo, Y. Liu, Y. Yang, C. Cheng, L. Li, J. Pi, Y. Si, H. Xiao, L. Li, S. Rao, F. Wang, J. Yu, J. Yu, D. Zou, P. Yi, The m6A reader YTHDF1 promotes ovarian cancer progression via augmenting EIF3C translation, Nucleic Acids Res, 48 (2020) 3816-3831.

[15] H. Shi, X. Wang, Z. Lu, B.S. Zhao, H. Ma, P.J. Hsu, C. Liu, C. He, YTHDF3 facilitates translation and decay of N(6)-methyladenosine-modified RNA, Cell Res, 27 (2017) 315-328.

Round 2

Reviewer 1 Report

No comments.

Author Response

There was no comments of reviewer. 
